# Bioinformatic Approach to Identify Potential *TGFB2*-Dependent and Independent Prognostic Biomarkers for Ovarian Cancers Treated with Taxol

**DOI:** 10.3390/ijms262411900

**Published:** 2025-12-10

**Authors:** Sanjive Qazi, Stephen Richardson, Mike Potts, Scott Myers, Saran Saund, Tapas De, Vuong Trieu

**Affiliations:** 1Oncotelic Therapeutics, 29397 Agoura Road, Suite 107, Agoura Hills, CA 91301, USA; stephen.richardson@oncotelic.com (S.R.); michael.potts@oncotelic.com (M.P.); scott.myers@oncotelic.com (S.M.); saran.saund@sapubio.com (S.S.); tde@oncotelic.com (T.D.); vtrieu@oncotelic.com (V.T.); 2Westmorland Campus, Kendal College, Market Place, Kendal LA9 4TN, UK

**Keywords:** biomarker, ovarian cancer, prognosis, transforming growth factors, tumor microenvironment, paclitaxel

## Abstract

High-grade serous ovarian carcinoma is the most common and aggressive form of ovarian cancer, accounting for over 60% of cases and nearly 75% of deaths, mainly due to late diagnosis and tumor aggressiveness. Standard treatment is platinum-based chemotherapy with paclitaxel, but relapse is frequent. This study aimed to identify prognostic biomarkers for patients with poor survival outcomes after Taxol treatment using bioinformatics analysis. We examined the effects of *TGFB2* mRNA expression and other markers on overall survival in serous ovarian cancer using the TCGA database, applying a multivariate Cox model that included interaction terms to identify *TGFB2*-dependent and independent prognostic markers, and controlling for age and treatment type. Candidate *TGFB2*-independent prognostic markers from TCGA were further validated using patient data from the KMplotter database. High *TGFB2* mRNA expression emerged as a prognostic biomarker for three potential gene targets (*TRPV4*, *STAU2*, and *HOXC4*) associated with improved OS at low levels of gene target expression, we identified four additional markers (*CLIC3*, *ANPEP*/*LAP1*, *RIN2*, and *EMP1*) that exhibited a *TGFB2*-independent negative correlation between mRNA expression and OS across the full spectrum of gene expression values in the ovarian cancer cohort validated using independent dataset from KMplotter, for Taxol-treated ovarian cancer patients. This study proposes a panel of potential prognostic biomarkers for the treatment of ovarian cancer patients, particularly by leveraging *TGFB2*-dependent mRNA expression as a significant biomarker, alongside four additional *TGFB2*-independent prognostic markers, for patients undergoing Taxol-based therapies. Future prospective clinical trials will be required to validate these prognostic markers.

## 1. Introduction

High-grade serous ovarian carcinoma (HGSOC) is the most common and lethal subtype of ovarian cancer, accounting for over 60% of epithelial ovarian cancers and nearly 75% of ovarian cancer deaths, mainly due to late-stage diagnosis and aggressive tumor biology [1]. Prognosis is difficult due to high recurrence rates, tumor heterogeneity, and variable responses to treatment, with most patients diagnosed at advanced stages and experiencing poor long-term survival [1,2,3]. HGSOC recurs in 70–80% of patients, with fewer than 15–30% surviving beyond five years, and only about 15% surviving ten years post-diagnosis [1,2,3,4,5].

Standard treatment involves cytoreductive surgery followed by intravenous platinum-based chemotherapy, typically carboplatin combined with paclitaxel, and may also include targeted agents such as bevacizumab or PARP inhibitors. While initial response rates are high (60–80%), most patients eventually experience relapse, often within three years, primarily due to the development of platinum resistance [3,6,7,8,9]. Long-term survivors are more likely to have undergone optimal cytoreductive surgery, have stage III (rather than IV) disease, and minimal residual disease after surgery. In addition, Breast Cancer (BRCA) gene status, molecular tumor subtype, and Homologous Recombination Deficiency (HRD) are associated with improved outcomes [4,5,9,10,11]. Homologous recombination gene mutations, especially BRCA1/2, confer increased sensitivity to platinum-based chemotherapy and PARP inhibitors, improving survival and progression-free survival (PFS) [8,10,12,13,14].

Our studies aimed to identify prognostic biomarkers using bioinformatics to identify patients with poorer survival outcomes when treated with Taxol. In future clinical trial design, these biomarkers could be used to identify patients responsive to Taxol delivered using nanoparticles, as exemplified by the use of paclitaxel-loaded PLGA-mPEG nanoparticles (NP-PTX) using TPGS (D-α-tocopheryl polyethylene glycol 1000 succinate) as a non-toxic emulsifier, thereby controlling the release of paclitaxel and demonstrating a significant reduction in acute toxicity [15].

One approach to biomarker discovery has been to utilize gene expression profiling models to facilitate risk stratification and predict therapeutic response [16,17]. These can then guide the use of targeted therapies, e.g., PARP inhibitors, and personalized treatment regimes, potentially improving survival and reducing overtreatment [10,11,14,16,18]. The seven-gene model predicts both overall survival (OS) and platinum sensitivity, enabling more tailored treatment approaches for patients with HGSOC [16,19].

Transforming Growth Factor Beta (TGF-β)-based (TGFB) mechanisms significantly influence the tumor microenvironment in ovarian cancers, thereby impacting tumor progression and metastasis and providing a potential target for biomarker discovery. Notably, TGF-β promotes tumor growth by interfering with antitumor T-cell immune responses and modifying components within the stroma and extracellular matrix, thereby facilitating immune evasion [20]. Additionally, TGF-β is associated with angiogenesis, a critical process for tumor survival and dissemination [21]. It also induces epithelial-to-mesenchymal transition (EMT) in ovarian cancer cells, which not only increases their metastatic potential but also alters cell adhesion, conferring migratory capabilities essential for cancer spread [22]. Furthermore, TGF-β enhances the invasion abilities of these cancer cells by upregulating matrix metalloproteinases (MMPs), facilitating the degradation of extracellular matrix components, and correlating with poorer clinical outcomes [23]. Importantly, elevated levels of TGF-β and its receptors in ovarian cancer tissues, plasma, and peritoneal fluid have been linked to adverse survival outcomes in patients [24]. Collectively, these findings underscore TGF-β’s pivotal role in fostering the aggressive characteristics of ovarian cancer by promoting tumor growth, invasion, and a tumor microenvironment conducive to cancer progression.

Our recent bioinformatics-led studies support the role of high levels of a specific TGF-β isoform, Transforming Growth Factor Beta 2 (*TGFB2*), and report that *TGFB2* gene methylation serves as a significant prognostic marker in adult glioblastoma (GBM) [25] and pancreatic adenocarcinoma (PDAC) [26,27]. In GBM, particularly among young adult males, elevated *TGFB2* methylation is associated with markedly improved OS compared to the well-established *MGMT* methylation marker [25]. Multivariate analyses confirm that *TGFB2* methylation is an independent predictor of improved OS, even after adjusting for *MGMT* and *TGFB1* methylation and clinical variables such as sex and age at diagnosis [25]. In PDAC, *TGFB2* methylation emerges as a positive prognostic marker, particularly in tumors with low CD8+ T-cell infiltration, suggesting an immunosuppressed environment. Patients with elevated *TGFB2* methylation and low expression of immune markers (such as *CD3D*, *LCK*, *HLA-DRA*) show median OS exceeding 50 months, representing a significant improvement in survival advantage [26]. Furthermore, the prognostic impact of *TGFB2* methylation is enhanced when considered alongside low immune marker expression, whereas high *TGFB2* mRNA levels in immunologically “cold” tumors correlate with poorer survival, underscoring the intricate relationship between methylation status and gene expression in determining patient outcomes in PDAC tumors [26].

Examination of *TGFB2* mRNA expression in pediatric brainstem diffuse midline glioma (pbDMG) tumors showed that high *TGFB2* mRNA levels were associated with an immunologically “cold” tumor microenvironment, characterized by low expression of antigen-presenting cell (APC) markers (e.g., *CD14*, *CD163*, *ITGAX/CD11c*), which impairs anti-tumor immune responses [28]. In PDAC, significant alterations in gene expression patterns were observed, particularly with a marked upregulation of *TGFB2* mRNA, which demonstrates a 7.9-fold increase over normal pancreatic tissue [29]. The prognostic implications of these expressions showed that high expressions of *TGFB2*, *IRF9*, or *IFI27* mRNA correlated with a poorer median overall survival (mOS), ranging from 16 to 20 months. In contrast, low expression of both *TGFB2* and either *IRF9* or *IFI27* mRNAs acted synergistically to be associated with markedly improved survival, with a mOS of 72 months [29]. In low-grade gliomas (LGG), elevated mRNA levels of *TGFB2* emerged as a significant negative prognostic marker, particularly when coupled with the activation of interferon-gamma receptor signaling via Interferon Regulatory Factor 5 (*IRF5*) and the increased expression of the immune checkpoint molecule B7-H3 [30]. Multivariate Cox regression models reveal that high *TGFB2* mRNA is an independent negative prognostic factor, with hazard ratios for OS ranging from 4.07 to 4.48, depending on whether *IFNGR2*, *STAT1*, *IRF1*, or *IRF5* is included [30].

We have extended these studies to utilize a bioinformatic-driven approach to characterize the impact of *TGFB2* mRNA, in combination with potential prognostic markers, on OS in ovarian cancer patients from The Cancer Genome Atlas (TCGA) database. We implemented a multivariate Cox proportional hazards model to directly compare hazard ratios (HRs) for *TGFB2* mRNA, a marker gene, while controlling for age at diagnosis and comparing patients who received chemotherapy-only (chemo-only) to those who received other therapies, including an interaction term between *TGFB2* and marker gene expression. Following the characterization of the impact of *TGFB2* mRNA-dependent screening, we sought to identify prognostic markers independent of *TGFB2* mRNA expression for patients who exhibited worse hazard ratios when treated with chemo-only. Using the prognostic marker list from the TCGA dataset, we cross-validated potential Taxol prognostic biomarkers in a larger cohort of patient data deposited in the KMplotter database. These prognostic markers will be used to stratify patients in future clinical trials employing nanoparticle-formulated Taxol-based therapies.

## 2. Results

### 2.1. TGFB2-Dependent Prognostic Marker Genes

Of the 15,984 marker genes screened, we prefiltered to identify 4904 genes with likelihood ratio test (LRT) *p*-values < 0.05 in the full Cox proportional hazards model assessing impact on OS, thereby identifying genes with favorable signal-to-noise ratios for detecting OS changes. Using this model, 808 genes significantly affected the *TGFB2*-dependent interaction term (*p* < 0.05 and HR for chemo-only > 1). For marker gene selection, we subsequently performed formal inference tests controlling for multiple comparisons using the Benjamini–Hochberg procedure (BH) (*Adjusted p*-*values* (*Adj. p*)). We compared the mRNA expression in tumor tissues of 709 Gene2 markers relative to normal tissues (*n* = 88), resulting in 110 significantly upregulated in tumor tissues (*Adj. p* < 0.001, fold increase > 1.5, log_2_ (TPM) expression in tumor tissue > 2) for 239 evaluable patients (Figure 1). We identified three clusters of gene expression profiles that showed a significant *TGFB2*-dependent impact on OS, with high expression (>4-fold increase) in tumor tissues compared to normal tissues. Eight genes exhibited greater than 340-fold increases in expression in tumor tissues (*CLDN6*, *KLK7*, *WNT10A*, *MYBL2*, *MAL2*, *KRT7*, *PRSS8*, *EPCAM*; Median (Range) of fold increase = 344 (261–7267)). Eighteen genes exhibited greater than 10-fold increase in expression (*LY6G6C*, *UPK3B*, *CRTAC1*, *KAZALD1*, *CDCA7*, *ANLN*, *SLC29A2*, *POU2F3*, *TJP3*, *KRT8*, *TMEM45B*, *KCNC3*, *PAQR4*, *PKMYT1*, *MMP15*, *TMEM184A*, *ERBB3*, *ARHGAP8*; Median (Range) of fold increase = 24.6 (11.9–98.8)). Eleven genes exhibited greater than 4-fold increases in tumor tissues (*RYR1*, *SPOCK2*, *ENTPD2*, *CITED4*, *MTHFD2*, *CDCA3*, *IDH2*, *SNTB1*, *PNP*, *PTPN6*, *MAP7*; Median (Range) of fold increase = 6.1 (4.3–8)) (Figure 1; Appendix A).

To identify prognostic markers based on 50th percentile cut-offs for the expression of *TGFB2* and the marker gene (Gene 2) and significant *TGFB2* by Gene2 interaction from the multivariate Cox proportional hazards model (Appendix A) and upregulated in tumor tissues (Table 1), we identified *TGFB2*/Gene2 combinations that exhibited significant curve separation between any two arms of the four survival curves (*Adj. p* < 0.01) (Figure 2, Appendix A). The Kaplan–Meier analysis highlighted *TRPV4* (Figure 2A), *MAL2* (Figure 2B), *STAU2* (Figure 2C), and *HOXC4* (Figure 2D), all of which showed a significant prognostic impact, with the effect dependent on *TGFB2* mRNA expression. The mOS for *TGFB2*^high^/*TRPV4*^low^ group of patients (*n* = 63) was 58.2 months compared to 38 months for *TGFB2*^high^/*TRPV4*^high^ group of patients (*n* = 58; *Adj. p* = 0.0011; mOS difference = −20.1 months) (Figure 2A). The mOS for the *TGFB2*^high^/*MAL2*^low^ group of patients (*n* = 65) was 35.9 months compared to 59.1 months for the *TGFB2*^high^/*MAL2*^high^ group of patients (*n* = 56; *Adj. p* = 0.0046; mOS difference = 23.3 months) (Figure 2B). The mOS for the *TGFB2*^high^/*STAU2*^low^ group of patients (*n* = 63) was 57.1 months compared to 35 months for the *TGFB2*^high^/*STAU2*^high^ group of patients (*n* = 58; *Adj. p* = 0.0068; mOS difference = −22.1 months) (Figure 2C). The mOS for the *TGFB2*^high^/*HOXC4*^low^ group of patients (*n* = 57) was 57.4 months compared to 38.2 months for the *TGFB2*^high^/*HOXC4*^high^ group of patients (*n* = 64; *Adj. p* = 0.0091; mOS difference = −19.2 months) (Figure 2D). At high levels of *TGFB2* mRNA expression, *TRPV4*, *STAU2*, and *HOXC4* were associated with worse survival outcomes, particularly at high levels of marker gene expression. High levels of *MAL2* mRNA expression were associated with improved survival at low levels of *TGFB2* mRNA expression.

We then examined the protein–protein interaction (PPI) network using STRING analysis to investigate the associations of *TGFB2* and the four marker genes with the expression of 110 genes that showed significant interaction effects between *TGFB2* and the marker genes in the Cox proportional hazards model (Appendix A). TGFB2 was found in the 19-node cluster with MAL2. Ovarian cancer patients with high *TGFB2* and *MAL2* mRNA levels showed the longest OS improvement. In contrast, low *TRPV4*, *STAU2*, and *HOXC4* expression exhibited improved survival outcomes at high *TGFB2* mRNA levels. PPI analysis showed TRPV4 formed its own 7-protein cluster, while STAU2 and HOXC4 did not associate with any of the eight clusters (Appendix A).

### 2.2. TGFB2-Independent Marker Genes That Predict Worse Prognosis in Taxol-Treated Patients

We next sought to identify prognostic markers independent of *TGFB2* mRNA expression for patients who exhibited worse hazard ratios when treated with chemo-only regimens, using the Cox multivariate proportional hazards model controlled for age at diagnosis, treatment (chemo-only versus all other treatments), and the interaction between *TGFB2* and the marker gene (Gene2). We first pre-screened the 15,984 marker genes using LRTs, identifying 4904 genes whose increased expression affected HR for the variables included in the Cox proportional hazards model relative to the null model (Model 1; LRT *p*-value < 0.05). We also prefiltered genes using parameters from the Cox proportional hazards model to identify genes significantly upregulated in tumor tissues. This pre-filter identified 841 genes with increased HR for Gene2 (*p* < 0.05, HR > 1, and HR for chemo-only > 1). To obtain a list of potential prognostic markers, we performed the hypothesis test corrected for multiple comparisons comparing the mRNA expression in tumor tissues of 841 Gene2 markers relative to normal tissues (*n* = 88), resulting in 100 significantly upregulated in tumor tissues (*Adj. p* < 0.001, fold increase > 1.5, Log_2_ (TPM) Expression in tumor tissue > 2) for 239 evaluable patients (Figure 3, Appendix A). We identified three clusters of gene expression profiles with higher expression in tumor tissues than in normal tissues. Three genes: *PRSS21*, *FOXQ1*, and *MMP7*, formed a highly upregulated cluster exhibiting a fold increase greater than 220-fold in tumor tissues (Median (Range) = 1273 (228–1598) fold increase). Thirteen genes showed greater than 9-fold increase in tumor tissues: *KIF1A*, *HPN*, *CRB2*, *SLC4A11*, *KLHL14*, *CLIC3*, *ADORA1*, *GGT6*, *ANO9*, *ARL4C*, *TMC4*, *LSR*, *SCNN1A* (Median (Range) = 26.4 (9.3–61.6) fold-increase). A cluster of twenty-one genes: *TGM1*, *SLC6A12*, *AHNAK2*, *RCOR2*, *STRA6*, *IL27RA*, *SPINT2*, *ZSWIM4*, *RASSF9*, *SLC7A1*, *GINS4*, *PODNL1*, *CILP2*, *TMEM119*, *VCAN*, *MXRA5*, *ANPEP*, *NTN1*, *MISP3*, *CD9*, *NINJ2*, exhibited greater than 3.5-fold increase in expression in tumor tissues (Median (Range) = 6.7 (3.5–13.1) fold-increase).

We cross-referenced the 100 genes identified in the TCGA dataset with those reported by Győrffy et al. 2023 [31,32], which significantly correlated OS with gene expression in all ovarian cancer patients using the best cut-off values, resulting in 30 overlapping genes (HR > 1, *Adj. p* < 0.05). We further narrowed the potential list of prognostic markers by applying more stringent criteria, comparing only median cutoff values for high- and low-expressing patient subgroups and selecting patients treated with Taxol. This analysis identified 12 significantly upregulated genes in ovarian cancer tumor tissues, which were cross-referenced in both the TCGA and KMplotter datasets (Figure 4). Two genes, *MISP3* (HR (95% CI range) = 1.64 (1.11–2.41); *p* = 0.012) and *TMEM38A* (HR (95% CI range) = 1.46 (1.23–1.74); *p* < 0.001), exhibited the most negative prognostic markers identified from the TCGA cohort of patients but not reported in the KMplotter dataset. Five genes exhibited low expression in normal tissue: *CLIC3*, *MISP3*, *ANPEP*, *ARL4C*, and *VCAN* (<2 TPM). Four of these genes were expressed greater than 10-fold in tumor tissue: *ANPEP* mRNA exhibited a significant (*Adj. p* < 0.0001) 10.2-fold increase in tumor (Mean ± SEM = 3.54 ± 0.13 log_2_TPM) compared to normal tissue (Mean ± SEM = 0.19 ± 0.18 log_2_TPM); *ARL4C* mRNA exhibited a significant (*Adj. p* < 0.0001) 26.4-fold increase in tumor (Mean ± SEM = 5.27 ± 0.1 log_2_TPM) compared to normal tissue (Mean ± SEM = 0.55 ± 0.14 log_2_TPM); *CLIC3* mRNA exhibited a significant (*Adj. p* < 0.0001) 19.3-fold increase in tumor (Mean ± SEM = 3.51 ± 0.12 log_2_TPM) compared to normal tissue (Mean ± SEM = −0.75 ± 0.2 log_2_RSEM); and *MISP3* mRNA exhibited a significant (*Adj. p* < 0.0001) 13.1-fold increase in tumor (Mean ± SEM = 3.35 ± 0.1 log_2_TPM) compared to normal tissue (Mean ± SEM = −0.36 ± 0.11 log_2_TPM) (Figure 4).

Using the prognostic marker list from the TCGA dataset, we cross-validated potential Taxol prognostic biomarkers in a larger cohort of patient data deposited in the KMplotter database (Figure 5).

Increasing *CLIC3* mRNA expression resulted in a significant increase in HR (HR (95% CI range) = 1.22 (1.03–1.45); *p* = 0.022) controlling for *TGFB2* mRNA expression (HR (95% CI range) = 0.95 (0.78–1.16); *p* = 0.622), age at diagnosis (HR (95% CI range) = 1.01 (1–1.03); *p* = 0.115), chemotherapy (HR (95% CI range) = 1.44 (0.95–2.19); *p* = 0.084) and interaction term (HR (95% CI range) = 0.95 (0.74–1.23); *p* = 0.715). Applying the multivariate Cox proportional hazards model parameters to patients with standard deviation units of −0.908, 0.943, 2.795, 4.647, and 6.498, respectively, yielded mOS of 48.3, 39.9, 33.9, 27.0, and 20.9 months (Figure 5A). There was a significant increase in HR for increasing levels of *ANPEP* mRNA expression (HR (95% CI range) = 1.45 (1.07–1.98); *p* = 0.017) controlling for *TGFB2* mRNA expression (HR (95% CI range) = 1.1 (0.87–1.41); *p* = 0.423), age at diagnosis (HR (95% CI range) = 1.02 (1–1.03); *p* = 0.053), chemotherapy (HR (95% CI range) = 1.42 (0.94–2.17); *p* = 0.099) and interaction term (HR (95% CI range) = 1.79 (0.96–3.32); *p* = 0.066). The figure shows the OS curves calculated using the multivariate Cox proportional hazards model parameters for patients with gene-expression levels of −0.373, 2.248, 4.869, 7.489, and 10.11 standard deviation units, with mOS of 44.9, 35.9, 20.9, and 17.4 months, respectively (Figure 5B). Increased levels of *RIN2* mRNA expression increased HR (HR (95% CI range) = 1.17 (1–1.36); *p* = 0.048), controlling for *TGFB2* mRNA expression (HR (95% CI range) = 0.94 (0.76–1.16); *p* = 0.555), age at diagnosis (HR (95% CI range) = 1.01 (1–1.03); *p* = 0.135), chemotherapy (HR (95% CI range) = 1.43 (0.94–2.16); *p* = 0.093) and the interaction term (HR (95% CI range) = 1.02 (0.84–1.24); *p* = 0.869). OS curves generated using multivariate Cox proportional hazards model parameters at gene-expression levels of −1.931, −0.479, 0.973, 2.425, and 3.877 standard deviation units exhibited median survival of 49.8, 44.9, 41.4, 36.3, and 34 months, respectively (Figure 5C). Increased levels of *EMP1* mRNA expression increased HR (HR (95% CI range) = 1.32 (1.12–1.56); *p* = 0.001) controlling for *TGFB2* mRNA expression (HR (95% CI range) = 1.01 (0.82–1.26); *p* = 0.905), age at diagnosis (HR (95% CI range) = 1.01 (1–1.03); *p* = 0.149), chemotherapy (HR (95% CI range) = 1.38 (0.91–2.09); *p* = 0.125) and the interaction term (HR (95% CI range) = 1.26 (0.93–1.7); *p* = 0.136). OS curves generated using the multivariate Cox proportional hazards model, with gene expression levels at −1.157, 0.784, 2.725, 4.665, and 6.606 standard deviation units, exhibited mOS of 51.4, 42, 35, 28.5, and 22.2 months, respectively (Figure 5D).

Evaluation of TCGA prognostic markers as prognostic biomarkers was conducted using the independent KMplotter dataset for patients who received Taxol treatment. The analysis focused on four genes (*CLIC3*, *ANPEP*/*LAP1*, *RIN2*, and *EMP1*), which were found to significantly influence OS in both the TCGA and KMplotter datasets (Table 1, Figure 6). Examination of the PPI network using STRING analysis for the associations of 100 genes with the four prognostic markers validated using the KMplotter database, which exhibited *TGFB2*-independent impact on OS (Appendix A). ANPEP appeared in the 23-node cluster. CLIC3 appeared in a 4-node cluster that was not connected to the main 23-node cluster. EMP1 and RIN2 showed no associations with any of the 8 clusters (Appendix A).

The availability of patient-level treatment metadata for the TCGA data set enabled us to investigate the impact of chemo-only further (Appendix A; Model 1), exposure to Taxol (Appendix A; Model 2), and patients receiving bevacizumab therapy (Appendix A; Model 3) using the multivariate Cox proportional hazards model. Examination of the Akaike information criterion (AIC) and LRT *p*-values showed no appreciable improvement in the fit of the additional Models 2 and 3 to the data (Appendix A; increases in AIC and LRT *p*-values were marginal and showed increases comparing Model 1 with Model 2, and comparing Model 1 with Model 3 (except for *RIN2*)). Model 1 provided the best fit to the data, and 171 of 191 (90%) patients in the chemo-only group were treated with Taxol (Appendix A). We then re-examined the Taxol-treated group (*n* = 171), patients treated with other chemotherapy agents (*n* = 20), and patients treated with other treatment modalities (*n* = 50) (Appendix A; Model 2). The application of Model 2 maintained the HR increases observed for all four prognostic markers (*p* < 0.05). Patients exposed to Taxol (*n* = 171) exhibited increases in HR that did not achieve statistical significance (HR ranged from 1.39 to 1.41). The cohort of patients exposed to bevacizumab (Appendix A; *n* = 29) showed improved survival (decreased HR), but the difference was not statistically significant.

The KMplotter data portal facilitated the investigation of patients’ stage and debulking status (Appendix A). In this analysis, patients presenting with stages 3 or 4 cancers (*n* = 720) showed significantly worse OS curves at high mRNA levels (median cut-offs) for all four prognostic markers (Appendix A; HR ranged from 1.27 to 1.51). Separation of OS curves was also observed for all four prognostic markers in patients undergoing optimal debulking (Appendix A; HRs ranged from 1.33 to 1.84), comparing patients with high versus low levels of the prognostic markers.

## 3. Discussion

### 3.1. TGFB2-Dependent Prognostic Markers

We classified gene expression profiles into three clusters based on genes that showed a significant *TGFB2*-dependent influence on OS, specifically those with high expression levels in tumor tissues (a 4-fold or greater increase relative to normal tissue).

Notably, eight genes showed expression levels exceeding 340-fold in tumor tissues, of which five have been implicated in ovarian cancers. High expression of *CLDN6* has been identified as an independent predictor of poor overall survival and PFS in ovarian cancer. It is thought to contribute to immune evasion, making it a promising therapeutic target [33,34]. *WNT10A* expression correlates with advanced disease stage, higher tumor grade, and significantly lower five-year survival rates, supporting its oncogenic function in ovarian cancer [35]. *MYBL2* is associated with increased cellular proliferation and migration, and its overexpression predicts aggressive disease and poor overall survival [36]. High levels of *KRT7* are associated with higher tumor grades, advanced stages, and poor prognoses, likely due to its role in promoting proliferation and EMT via the TGF-beta/SMAD2/3 signaling pathway [37]. *PRSS8* is also overexpressed in ovarian cancer and may contribute to tumor progression and EMT, with its prognostic value possibly varying by cancer subtype; it is considered a potential biomarker for early detection and disease monitoring [38].

Eighteen genes had expression changes greater than 10-fold. Patients with high *ERBB3* expression tend to have shorter median survival times compared to those with lower *ERBB3* levels [39]. In a study that combined preclinical and clinical evidence, researchers investigated aberrant ERBB3 activity that contributes to the development of resistance and/or treatment failure. Several ERBB3-directed monoclonal antibodies, bispecific antibodies, and emerging antibody–drug conjugates demonstrated encouraging clinical outcomes that enhance therapeutic efficacy and address resistance, particularly when combined with other anti-cancer strategies [40]. ARHGAP8, or BPGAP1, is a RhoGAP protein linked to cancer progression. In ovarian cancer, it is often upregulated, promoting cell migration and proliferation, which may lead to a poorer prognosis for patients [41]. High *CDCA7* expression is associated with poorer survival outcomes, including overall survival and PFS [42].

Eleven genes showed more than a 4-fold increase in expression in tumor tissues. High expression of the *SPOCK2* gene in ovarian cancer is associated with a poorer prognosis and reduced overall survival. Specifically, *SPOCK2* promotes the invasion and migration of ovarian cancer cells, potentially through interactions with *ITGA3* and the activation of FAK signaling, and is associated with a higher likelihood of metastasis [43]. *MAP7* also plays a role in cisplatin resistance in ovarian cancer by modulating the Wnt/β-catenin pathway [44].

Kaplan–Meier analysis showed that, at high levels of *TGFB2* mRNA expression, *TRPV4*, *STAU2*, and *HOXC4* were associated with poorer survival outcomes, particularly at high marker gene expression levels. In contrast, high *MAL2* mRNA expression was associated with improved survival, whereas high *TGFB2* mRNA expression was associated with poorer survival.

Transient Receptor Potential Vanilloid 4 (TRPV4) is a non-selective cation channel widely expressed in human tissues. It primarily functions as a polymodal sensor, responding to mechanical, osmotic, and thermal stimuli, and regulates calcium influx, which is crucial for numerous cellular processes [45,46]. *TRPV4* is highly expressed in ovarian cancer and is associated with poor prognosis, specifically worse overall survival (OS), disease-specific survival (DSS), disease-free interval (DFI), and PFS in ovarian cancer [47,48]. It can promote proliferation and migration [49] and is considered an oncogene and a prognostic marker in ovarian cancer [48,49]. The oncogenic pathway of *TRPV4* in ovarian cancer may be related to fatty acid synthesis, through the calcium-mTOR/SREBP1 signaling pathway, thereby promoting ovarian cancer progression [48]. *TRPV4* expression is also closely associated with immune regulation-related pathways. Tumor-associated macrophage infiltration levels are positively correlated with *TRPV4* expression in TCGA pan-cancer samples [48]. It has been suggested that patients with high expression of *TRPV4* might also be more resistant to the treatment of cisplatin and oxaliplatin [48]. *TRPV4* is therefore a potential therapeutic target for ovarian cancer [47]. Additional research is required to confirm this target through protein-level analyses.

Myelin and Lymphocyte protein 2 (MAL2) is a tetra-transmembrane protein belonging to the MAL proteolipid family. It is widely expressed in various human tissues and plays key roles in intracellular membrane trafficking, especially in polarized epithelial cells [50]. *MAL2* was found to be significantly overexpressed in high-grade serous carcinomas compared with serous borderline tumors [51,52] expression was highest in serous carcinomas relative to other histological subtypes but was not found to be correlated to patient survival [51,52]. More recently, it was reported that *MAL2* was dysregulated in multiple cancers and was related to patient overall survival (OS), mutation, and drug sensitivity. Furthermore, experimental results showed that *MAL2* deletion negatively regulated proliferation, migration, and invasion of ovarian cancer [53]. The authors suggested *MAL2* could be a novel oncogene that can activate EMT [53]. Our results suggest that patients with high *MAL2* and *TGFB2* expression have improved survival and warrant further validation through protein-level analysis.

PPI analysis revealed that TGFB2 and MAL2 were associated via a 19-node cluster of proteins using a 110-gene set. This 19-node cluster showed that TGFB2 has first-level connections to ICAM1, ITGB8, FGFR2, and ERBB3. Interestingly, ICAM1 was found to activate EMT via TGF-beta signalling in triple-negative breast cancers [54]. TGF-β primes FGFR isoform switching and generates fibroblastic cells by EMT in NMuMG cells; an epithelial-like cell isolated from the mammary gland of a mouse [55]. In mouse models of melanoma and breast cancer, regulatory T cells (Tregs) expressing Itgβ8 were identified as the primary tumour cells that activate TGF-β. When Itgβ8 is deleted from Treg cells, TGFβ signalling is disrupted in T lymphocytes within the tumour, resulting in more effective tumour suppression [56]. In HER2-overexpressing mammary epithelial cells, TGF-β activates the PI3K/Akt pathway, boosting cell survival and migration [57]. This study demonstrated that TGF-β exposure induced phosphorylation and cell-surface relocation of TACE/ADAM17, which, in turn, increased growth factor release, thereby stimulating PI3K/Akt activation via enhanced p85–ErbB3 interaction. Blocking *TACE* or ErbB3 with RNA interference halts TGF-β-driven Akt activation and invasiveness [57]. These associations will require validation in ovarian cancer as the PPI observations are correlative.

Staufen double-stranded RNA binding protein 2 (STAU2) is an RNA-binding protein with critical roles in post-transcriptional gene regulation. Its functions are prominent in the nervous system and extend to other cell types and developmental processes [58,59,60]. In recently published research on PDAC, high *STAU2* expression has been linked to tumor progression, immune evasion, and chemotherapy resistance [61], and experimental studies indicated that *STAU2* promoted aggressive tumor behavior. The authors suggested that *STAU2* could be a novel prognostic and diagnostic biomarker for PDAC [61]. Our studies will require validation in ovarian tumors through inclusion in prospective clinical trials.

*HOXC4* is a member of the homeobox (HOX) gene family, which encodes transcription factors crucial for regulating gene expression during embryonic development and maintaining tissue identity in adult organisms [62]. Ovarian cancer-derived cell-line SK-OV3, but not OV-90, exhibited highly dysregulated expression of members of the *HOX* gene family, disrupting the interaction between HOX proteins and their co-factor PBX induced apoptosis in SK-OV3 cells, and retarded tumour growth in vivo [63]. A review concluded that *HOX* genes contribute to the oncogenesis of ovarian cancer by inhibiting apoptosis, promoting DNA repair, and enhancing cell motility. Additionally, it suggested that *HOX* genes have potential roles as prognostic and diagnostic markers, as well as therapeutic targets, in this disease [64]. In terms of identifying the role of specific *HOX* genes in ovarian cancer, subsequent research found that 36 of the 39 *HOX* genes were overexpressed in high-grade serous epithelial ovarian cancer compared with normal tissue [65]. The authors also determined that *HOXA13*, B6, C13, D1, and D13 were prognostic of poor clinical outcomes [65]. A later review of *HOX* genes in high-grade ovarian cancer concluded that *HOX* genes show a highly dysregulated expression in ovarian cancers of all histological subtypes, with many studies showing either up- or down-regulation of *HOX* genes [66]. A more recent review identified *HOXC4* expression across 21 tumor cell lines and found that it was significantly higher than in normal tissues across 21 tumor types. The survival analysis revealed that *HOXC4* upregulation in several cancers was associated with a worse prognosis. *HOXC4* was observed to correlate closely with colon adenocarcinoma (COAD), head and neck squamous cell carcinoma (HNSC), LGG, liver hepatocellular carcinoma (LIHC), rectum adenocarcinoma (READ), and thyroid carcinoma (THCA) in terms of tumor immune cells infiltration [67].

### 3.2. TGFB2-Independent Prognostic Markers

We identified three potential targets (*TRPV4*, *STAU2*, and *HOXC4*) to improve OS outcomes for patients with high *TGFB2* mRNA levels. We found that the combination of high *TGFB2* and high target gene expression was associated with worse OS than high *TGFB2* and low target gene expression. To expand the list of potential biomarkers for ovarian cancer patients, we sought to identify biomarkers independent of *TGFB2* mRNA expression using a Cox proportional hazards model. Examination of this gene list, comparing expression in tumor versus normal tissues, revealed three clusters of genes that were significantly upregulated in tumors. We identified three clusters of gene expression profiles with higher expression in tumor tissues than in normal tissues.

Three genes (*PRSS21*, *FOXQ1*, and *MMP7*) formed a highly upregulated cluster, with a fold increase greater than 220 in tumor tissues. The *PRSS21* gene, which encodes protein testisin, is associated with ovarian cancer prognosis. Specifically, increased testisin expression, particularly in primary tumors, can inhibit tumor metastasis and ascites accumulation, leading to better outcomes. However, in advanced serous carcinomas, testisin expression may be decreased compared to primary tumors, potentially contributing to poorer prognosis [68]. In ovarian cancer, the *FOXQ1* gene is associated with a poorer prognosis, meaning higher levels of *FOXQ1* expression often correlate with a worse outcome for patients. Studies have shown that *FOXQ1* promotes tumor progression in ovarian cancer both in vitro and in vivo. Specifically, *FOXQ1* appears to mediate this effect by influencing the WNT/β-catenin signaling pathway [69]. MMP-7 facilitated the invasion and migration of ovarian tumor cells, indicating its key function in ovarian cancer progression. Immunohistochemistry analysis demonstrated abnormally increased Gli2 and MMP-7 expression levels in benign tumors and ovarian cancer tissues. Moreover, high MMP-7 levels were significantly associated with poor overall survival (OS) and poor PFS in ovarian cancer patients [70].

Thirteen genes showed greater than 9-fold increase in tumor tissues, of which four have been previously reported to have exerted prognostic impact in ovarian cancers. High expression of the *KIF1A* gene in ovarian cancer is associated with a poorer prognosis, including shorter overall survival and post-progression survival. Specifically, studies have shown that ovarian cancer patients with higher *KIF1A* expression have a significantly reduced overall survival rate compared to those with lower expression. This suggests that *KIF1A* could be a potential prognostic biomarker for ovarian cancer [71]. High expression of the *SLC4A11* gene is associated with poorer overall survival in patients with ovarian cancer, particularly in those with HGSOC. This means that ovarian cancer patients with higher levels of *SLC4A11* tend to have a lower chance of surviving their cancer [72]. In ovarian cancer, the increased expression of the *KLHL14* gene is associated with a poorer prognosis. Studies using databases like TCGA have shown that higher *KLHL14* levels correlate with reduced overall survival. Further analysis, including pathway analysis and immune cell infiltration studies, suggests that *KLHL14* plays a role in the tumor microenvironment and has potential as a therapeutic target [73]. The *ADORA1* gene, which encodes the adenosine A1 receptor, has been investigated for its role in the prognosis of ovarian cancer. Studies suggest that high *ADORA1* expression is associated with poorer OS in ovarian cancer patients. This correlation is also observed with recurrence-free survival [74]. High expression of the *LSR* (lipolysis-stimulated lipoprotein receptor) gene in epithelial ovarian cancer (EOC) is associated with a poorer prognosis for patients. LSR is a protein that appears to promote EOC cell survival, particularly under conditions of glucose deprivation. Furthermore, LSR is implicated in tumor growth and metastasis [75]. In ovarian cancer, *SCNN1A* expression is associated with patient prognosis. Specifically, high expression of *SCNN1A* is associated with poorer overall survival and PFS in ovarian cancer patients. This suggests *SCNN1A* could be a potential biomarker for prognosis in ovarian cancer [76].

A cluster of 21 genes showed a greater than 3-fold increase in expression in tumor tissues. High expression of the *SLC6A12* gene in ovarian cancer is associated with a poorer prognosis, specifically for serous-type ovarian cancer. This suggests that *SLC6A12* could be a potential target for therapies to improve patient survival [77]. Solute carrier (SLC) transporters correlate with poorer survival rates in ovarian cancer patients [78]. A study found that the expression of *IRF1* and *STRA6* was markedly upregulated in the ovarian cancer cell line HEY. *STRA6* markedly decreased the invasion and migration ability of the ovarian cancer cell line HEY [79]. In HGSOC, *SPINT2* is upregulated and associated with a shorter survival time and poorer prognosis. This indicates a potential tumor-promoting role in this specific type of ovarian cancer [80]. *ZSWIM4*, a gene, is associated with poorer prognosis in epithelial ovarian cancer due to its overexpression in tumor tissues and its correlation with increased recurrence rates. Furthermore, *ZSWIM4* overexpression has been linked to chemotherapy resistance in EOC cells. Specifically, *ZSWIM4* expression is upregulated following carboplatin treatment, a standard chemotherapy drug, contributing to chemoresistance [81]. Studies have shown that high *PODNL1* expression is associated with poorer overall survival in ovarian cancer [82]. *TMEM119* overexpression in ovarian cancer is associated with poorer prognosis and promotes cancer progression. Studies have shown that *TMEM119* facilitates the proliferation, invasion, and migration of ovarian cancer cells, potentially through the PDGFRB/PI3K/AKT signaling pathway [83]. High *VCAN* expression, particularly in the stromal tissue surrounding the tumor, is linked to a worse prognosis for ovarian cancer patients [84].

### 3.3. Taxol-Treatment Prognostic Biomarkers

We conducted a cross-referential analysis of the 100 genes identified within the TCGA dataset (90% of the chemo-only cohort of patients were treated with Taxol, and evaluated using the multivariate Cox proportional hazards model) alongside the findings of Győrffy et al. (2023) [31,32], who established significant correlations OS and gene expression among all ovarian cancer patients, utilizing optimal cut-off values. To refine our candidate list of prognostic markers, we applied more stringent criteria, specifically focusing on median cut-off values to distinguish between high- and low-expressing Taxol-treated patient sub-groups.

Recent studies have identified potential mechanisms for the development of paclitaxel resistance involving TGFB pathways in ovarian cancer cell lines via [85] and cyclin dependent kinase 14 (*CDK14*) [86] mediated mechanisms. We applied a multivariate Cox proportional hazards model to investigate the impact of Taxol-treated patients on HR for the prognostic markers identified in our screen. These analyses showed that statistical significance of the Taxol parameter could not be achieved and that would have warranted further characterization of paclitaxel-sensitive versus paclitaxel-resistant tumors in the TCGA dataset. Likewise, we also tested the impact of bevacizumab on HR in ovarian cancer patients in the TCGA dataset because of the recent introduction of bevacizumab into first-line therapeutic regimens [87] and improvements observed in PFS during second-line maintenance [88]. We observed improvements in OS for 29 patients treated with bevacizumab that did not achieve statistical significance, so we could not further investigate the prognostic impact of the markers described in our study.

Our analysis focused on four genes (*CLIC3*, *ANPEP*/*LAP1*, *RIN2*, and *EMP1*). CLIC3 is a multifunctional protein involved in the progression, metastasis, and therapy resistance of several cancer types. CLIC3 has been identified as critical for cell migration and invasion in PDAC, where it regulates cell rear detachment and sustains Src signaling in 3D environments [89]. Its expression in PDAC predicts poor prognosis and lymph node metastasis, suggesting an early and direct link to tumor aggressiveness and progression [89]. Mechanistically, CLIC3 acts in concert with Rab25 to recycle α5β1 integrin, facilitating cancer cell invasiveness by promoting integrin’s trafficking from late endosomes to the plasma membrane [90]. CLIC3 is highly abundant in cancer-associated fibroblasts (CAFs), where it is secreted into the extracellular matrix (ECM). In CAFs and tumor stroma, CLIC3 enhances ECM stiffness and angiogenesis in collaboration with the enzyme transglutaminase 2 (TGM2), driving the pro-invasive and pro-angiogenic functions of the tumor microenvironment. CLIC3’s extracellular activity as a glutathione-dependent oxidoreductase activates TGM2, thereby amplifying processes such as basement membrane disruption and endothelial cell invasion [91,92,93]. *CLIC3* is also highly expressed in the stroma of aggressive ovarian tumors and is a prominent component of the CAF secretome. Preliminary tissue microarray analysis demonstrates strong *CLIC3* staining in the stroma of highly vascularized ovarian cancers, further supporting its association with tumor aggressiveness. Elevated levels of *CLIC3* in ovarian cancer tissue, both in the tumor cell and stromal compartments, correlate with poor clinical outcomes [91,92,93]. Furthermore, high *CLIC3* expression is associated with increased resistance to cisplatin in ovarian cancer (OC), which is mechanistically underpinned by enhanced integrin β1 redistribution and activation of the PI3K-AKT pathway, thereby contributing to chemoresistance and metastatic competence [94]. CLIC3 enzymatic activity makes it a promising therapeutic target. The GSH-dependent oxidoreductase activity, especially in the extracellular milieu, can potentially be inhibited pharmacologically, as evidenced by early identification of small-molecule CLIC3 inhibitors with anti-migratory effects in preclinical models [91]. Its overexpression in a wide range of cancers, compared with normal tissues, and its detectability in interstitial fluids, support its candidacy as a diagnostic and prognostic biomarker [95,96,97,98,99]. PPI analysis showed that CLIC3 was associated with VSIG4 which is overexpressed in ovarian tumors exhibiting with progression and recurrence of the cancer [100].

ANPEP (aminopeptidase N/CD13) is a multifunctional enzyme with significant roles in cancer biology, influencing tumor progression, angiogenesis, immune modulation, and therapeutic resistance. ANPEP promotes tumor vascularization and metastasis through its enzymatic activity and non-enzymatic interactions. CD13+ myeloid cells and stromal cells work together to enhance angiogenesis, thereby facilitating tumor growth and metastasis [101,102]. ANPEP affects cytokine production, T-cell activity, and macrophage function, contributing to an immunosuppressive tumor microenvironment [103]. ANPEP marks semi-quiescent CSCs, which resist chemotherapy and drive relapses by reducing ROS-induced DNA damage [104]. In prostate cancer, overexpression in African American (AA) patients correlates with inflammatory macrophage infiltration, cholesterol transport, and androgen signaling, potentially explaining worse outcomes in AA men [105]. However, reduced *ANPEP* expression and hypermethylation in prostate cancer tissues are linked to recurrence risk [106]. In liver cancer, CD13+ CSCs survive genotoxic stress (e.g., 5-FU), and combining CD13 inhibitors with ROS-inducing therapies reduces tumor volume in preclinical models [104]. In PDAC, elevated serum ANPEP outperforms CA19-9 in early-stage detection and correlates with advanced TNM (Tumor Node Metastasis) stage and shorter survival [107]. In lung cancer, CD13 expression in non-small-cell lung cancer (NSCLC) is associated with lymph node metastasis, advanced stage, and reduced survival [108]. In ovarian cancer, ANPEP enhances tumor motility, MMP-2/VEGF secretion, and metastasis. Inhibitors demonstrate efficacy when used early; however, overexpression may paradoxically reduce cisplatin sensitivity [109,110,111]. *ANPEP* thus exhibits context-dependent roles (e.g., pro-tumorigenic in most cancers but tumor-suppressive in specific ovarian cancer models). Inhibitors (e.g., bestatin) and antibody-based strategies show promise in preclinical studies, particularly when combined with chemotherapy or radiation. Serum ANPEP levels and tissue expression patterns could guide diagnosis, prognosis, and treatment stratification. PPI analysis revealed that ANPEP was present in the 23-node cluster and showed first-level associations with GGT6, CD9, TGFBI, and GAPDH. Transforming Growth Factor Beta-Induced Protein (*TGFBI*) expression is elevated in platelet-treated ovarian cancer cells and is associated with poorer patient outcomes. It serves as an independent predictor of poor prognosis, promoting migration and invasion by regulating EMT markers and ECM-degrading proteins through PI3K/Akt activation with integrin αvβ3 [112].

*RIN2* is crucial for regulating endosomal trafficking, a process that has significant implications for cancer-related signaling pathways and receptor trafficking [113,114]. *RIN2* plays a role in angiogenesis, which is a vital process for tumor growth and metastasis [114]. The role of *RIN2* in cell adhesion may contribute to cancer metastasis [113]. Recent research has also implicated *RIN2* in the progression of triple-negative breast cancer [115].

Loss of *EMP1* in these contexts is associated with higher tumor grade, advanced stage, and poor prognosis [116,117]. In colorectal cancer, low *EMP1* expression significantly correlates with advanced T stage, lymph node metastasis, clinical stage, and poor five-year survival [118]. Overexpression of *EMP1* leads to increased apoptosis via caspase-9, reduced VEGF-C expression, and diminished cancer cell ‘stemness’ [118,119]. In contrast, *EMP1* is upregulated in GBM, melanoma, leukemia, NSCLC, bladder cancer, and some ovarian cancers, where it promotes tumor cell migration, invasion, and metastasis [117,120]. High *EMP1* expression in these cancers is often associated with advanced stage and poor overall survival [120,121]. *EMP1* enhances metastasis through pathways such as copine-III/Src/Vav2/Rac1 and PI3K/AKT, and is involved in regulating tight junctions, immune cell infiltration, and drug resistance [117]. *EMP1* expression correlates with immune cell infiltration (macrophages, neutrophils, dendritic cells) and high expression of immune checkpoints in the tumor microenvironment, especially in bladder and ovarian cancers [120,121]. In ovarian cancer, *EMP1* is frequently upregulated, especially in advanced and chemoresistant cases, and high *EMP1* expression is associated with poor prognosis, advanced stage, metastasis, and cisplatin resistance [120,122,123,124]. EMP1 promotes proliferation, invasion, and EMT, via the MAPK (RAS/RAF/MAPK/c-JUN) pathway [120,122,123]. High *EMP1* expression also correlates with immune cell infiltration and immune checkpoint expression [120].

An assessment of the prognostic markers identified in TCGA as potential prognostic biomarkers was performed using the independent KMplotter dataset, which comprises patients treated with Taxol. This analysis showed that four genes: *CLIC3*, *ANPEP*/*LAP1*, *RIN2*, and *EMP1*, were found to exert a significant impact on overall survival (OS) in both the TCGA and KMplotter cohorts [3,6,7,8,9].

### 3.4. Future Clinical Design of Taxol-Prognostic Biomarkers in Combination with Lipid-Based Nanoparticles of Taxol

Taxol/paclitaxel is a potent chemotherapeutic agent; however, it is also hydrophobic, and consequently, it is often formulated with Cremophor EL, a surfactant known to cause hypersensitivity reactions in some patients [15]. Lipid-based nanoparticles (LNPs) have emerged as effective vehicles for enhancing Taxol delivery by improving drug solubility, targeting specificity, and therapeutic efficacy, while simultaneously minimizing toxicity. For example, Lv et al. (2015) [15] developed paclitaxel-loaded PLGA-mPEG nanoparticles (NP-PTX) using TPGS (D-α-tocopheryl polyethylene glycol 1000 succinate) as a non-toxic emulsifier. This formulation eliminated the need for Cremophor EL and achieved encapsulation efficiencies exceeding 90%. The resulting nanoparticles provided controlled paclitaxel release and demonstrated a significant reduction in acute toxicity. Zhai et al. (2018) [125] developed two types of self-assembled lipid nanoparticles for delivering paclitaxel: non-targeted (PTX-CB) and EGFR-targeted (EGFR-PTX-CB) nanoparticles, aiming to enhance solubility and reduce toxicity. This approach also improved treatment effectiveness, as lipid nanoparticles enabled more efficient drug loading and enhanced cytotoxicity against cancer cells. Khalifa et al. (2019) reviewed strategies for various paclitaxel-loaded nano-delivery systems for ovarian carcinoma. They reported that nanosuspensions overcome the solubility challenges associated with hydrophobic drugs such as paclitaxel and improve bioavailability, as nanosuspensions can extend circulation times in the bloodstream, thereby increasing accumulation in tumors [126]. Our research program is developing LNPs to deliver compounds such as Taxol to tumors, and our bioinformatics-led approach has identified a potential biomarker panel to select patients for treatment with Taxol formulated in nanosuspensions.

The KMplotter dataset did not provide patient-level treatment information thereby presenting our study with a limitation of insufficient treatment regimen matching, which will require future head-to-head clinical trials for validation. This study is in the bioinformatics screening stage, generating hypotheses for further testing; due to limited access to clinical samples, Reverse Transcription Polymerase Chain Reaction (qRT-PCR)/Immunohistochemistry (IHC) validation has not been conducted. In future work, we will collect clinical samples from paclitaxel-treated ovarian cancer patients to validate protein expression levels of the biomarkers and the consistency between mRNA and protein expression. The *TGFB2*-dependency will require additional measurements of mRNA levels of *TGFB2* along with the prognostic markers in the cellular compartments of the tumor. Thus, a validation pipeline combining qRT-PCR, IHC, and single-cell RNA-seq studies will enable comprehensive confirmation of biomarkers by integrating bulk quantitative mRNA measurement, spatial protein localization, and single-cell resolution of tumor heterogeneity. Additionally, preclinical studies will be required to evaluate the bioequivalence of the pharmacokinetic and efficacy profiles of LNP-formulated versus unformulated Taxol in patients with high levels of *TGFB2* and the companion markers, as well as the *TGFB2*-independent prognostic markers.

## 4. Materials and Methods

### 4.1. AI-Augmented Summaries of Pubmed Abstracts

Pubmed searches using the keywords: “Gene AND TCGA AND Prognosis AND Ovarian retrieved 787 abstracts”; “Ovarian AND single-cell RNAseq” (767 abstracts); and “Ovarian AND TGF-beta” (1718 abstracts) (https://pubmed.ncbi.nlm.nih.gov/, accessed on 17 December 2024) were downloaded as text documents for processing using the Oncotelic Chatbot technologies (see Appendix A). Chatbot technology enables users to conduct question-and-answer sessions to refine references for manuscript preparation. This was not a systematic review, but an expansion of keywords used for PubMed searches by having an interactive console to ask further questions to refine the screen of published articles. The chatbots were not used for any validation of the results, for formulating hypotheses, or for writing the paper. The bioinformatics analyses identified all biomarkers reported in the manuscript.

### 4.2. Using the Multivariate Cox Proportional Hazards Model to Identify TGFB2-Dependent and TGFB2-Independent Biomarkers Impacting Overall Survival (OS) Outcomes for Serous Ovarian Cancer Patients

We analyzed clinical metadata (https://www.cbioportal.org/study/summary?id=ov_tcga_pan_can_atlas_2018; accessed on 28 November 2023) and RNA sequencing data (“data_mrna_seq_v2_rsem.txt”: Batch normalized from Illumina HiSeq_RNASeqV2 using the RSEM algorithm (Transcripts per million (TPM)) from 241 patients diagnosed with Ovarian cancer to compare OS correlates to mRNA expression values.

Patient-level treatment information (treatment regimen reported as “TREATMENT_TYPE”, “AGENT”, “START_DATE”, “STOP_DATE”, “NUMBER_OF_CYCLES”) was available for the ovarian cancer dataset (*n* = 241) that was utilized to stratify patients based on exposure to chemotherapy (*n* = 238; with most common treatment regimens containing carboplatin (*n* = 219), paclitaxel (*n* = 215), doxorubicin (*n* = 80), cisplatin (*n* = 72), and topotecan (*n* = 68)); targeted molecular therapy (most common being bevacizumab (*n* = 29); radiation therapy (*n* = 17); hormone therapy (most common being Tamoxifen (*n* = 17)); immunotherapy (*n* = 5). The chemotherapy-treated group of patients was further subdivided into two groups: chemo-only for patients that were not exposed to targeted molecular therapy, radiation therapy, hormone therapy, or immunotherapy (*n* = 191); and others (*n* = 50). We also determined the number of patients that were exposed to paclitaxel from the chemo-only group (Taxol; *n* = 170), patients exposed to chemotherapy but not paclitaxel or other treatment regimens (other chemo-only; *n* = 20), and others (*n* = 50).

Multivariate analyses were carried out using the Cox proportional hazards model to assess the individual effects of *TGFB2* and Gene2 mRNA expression levels (Screened 15,984 genes) on OS, and to calculate HR. This analysis was controlled for age at diagnosis, treatment (chemo-only versus all other treatments), and the interaction between *TGFB2* and Gene2 (Model 1). Briefly, the model included (i) *TGFB2* mRNA levels as a linear covariate expressed as Zscores (*n* = 241), (ii) The mRNA expression level for Gene2 mRNA levels as a linear covariate expressed as Zscores (*n* = 241), (iii) *TGFB2* by Gene2 interaction term to determine the dependency of Gene2 prognostic impact on *TGFB2* mRNA levels (*n* = 241), (iv). Treatment comparing chemo-only (*n* = 191) versus other (*n* = 50) treatment regimens, and (v). Age at diagnosis expressed as a linear covariate (*n* = 241), implemented in R (survival_3.2-13 ran in R version 4.1.2. Forest Plots were used to visualize hazard ratios from Cox proportional hazards models for OS outcomes (survminer_0.4.9 ran in R version 4.1.2 (1 November 2021). The life table HRs were estimated using the exponentiated regression coefficient from Cox proportional hazards analyses implemented in R (survival_3.2-13, running in R version 4.1.2).

Prefiltering was performed by selecting genes with a mean TPM greater than 10 across all ovarian cancer patient samples for further analysis. Potential prognostic marker genes were identified using LRTs, which were used to filter and preselect genes before conducting further statistical analyses, including comparisons of normal versus tumor expression and Kaplan–Meier survival analyses. Specifically, the LRT compared two nested models: a null model, in which all coefficients were set to zero, and the dependent variables were assumed not to affect the HR, and an alternative model that included all dependent variables in the regression. The maximized log-likelihoods from both models were used to compute *p*-values, thereby assessing whether including the dependent variables significantly improved model fit. For downstream analysis, we retained marker genes with *p*-values less than 0.05. This filtering approach increased the statistical power of subsequent formal tests by narrowing the number of hypotheses considered before multiple-testing adjustments of *p*-values for differential expressions in normal versus tumor comparisons.

Following the screening of Cox proportional hazards models using chemo-only as a variable (Model 1), we investigated two additional Cox proportional hazards models, whereby in the first model the treatment factor was further stratified to Taxol (*n* = 171 patients), other chemo-only (*n* = 20 patients), and other therapies (*n* = 50) (Model 2). The second additional model included bevacizumab as an additional factor in Model 1 (Model 3). The Akaike Information Criterion (AIC) was used to compare different Cox proportional hazards models for the prognostic marker genes pre-screened using Model 1. We tested whether the Cox proportional hazards model with the best fit, comparing Model 1 (chemo-only) versus Model 2 (Taxol exposure), and Model 1 (chemo-only) versus Model 3 (bevacizumab inclusion), improved fit by lowering the AIC and LRT *p*-values. Models 2 and 3 tested the Taxol-specific effect and whether any multicollinearity impacts the main Gene2 effect. The AIC values from a fitted coxph object in R were extracted from the base stats::AIC functions (stats library version 4.4.2). The HR estimate for the prognostic marker gene was reported from the Cox proportional hazards model with the lowest AIC.

To illustrate how different combinations of *TGFB2* mRNA levels and Gene2 mRNA expression affected OS, we generated predicted survival proportions at any given time by plotting and calculating the shift in the baseline OS curve for 241 ovarian cancer patients (138 death events) using parameters from the output of the multivariate regression model and then utilizing the fitted parameters to shift the baseline curves using values fixed for the control variables and varying a single variable of interest, thereby evaluating the *TGFB2* mRNA dependent impact of the prognostic marker gene on OS. The shift in the baseline survival curves represented the predicted median OS times throughout the entire range of marker gene expression levels (Minimum, Lower quartile, Median, Upper Quartile, and Maximum), and the magnitude of the HR calculation.

### 4.3. Differential Expression of mRNA Comparing Serous Ovarian Tumors Versus Normal Tissue Samples

To identify genes enriched in tumor samples, we compared mRNA expressions in tumors compared to normal tissue samples. We utilized log_2_-transformed transcripts per million (TPM) summarized RNAseq datafiles (https://toil-xena-hub.s3.us-east-1.amazonaws.com/download/TcgaTargetGtex_rsem_gene_tpm.gz accessed on 7 December 2025; Full metadata) downloaded from the UCSC Xena web platform (https://xenabrowser.net/datapages/ accessed on 25 July 2023) [127] to compare gene expression levels for 88 evaluable Normal Ovarian samples (“GTEX Ovary”) versus 239 Ovarian cancer patients (“TCGA Ovarian Serous Cystadenocarcinoma”). This resource reports results from the UCSC Toil RNAseq recompute compendium, which is a standardized, realigned and recalculated gene and transcript expression data set for all TCGA, TARGET, and GTEx samples [128] that enables users to contrast gene and transcript expression between TCGA “tumor” samples and corresponding GTEx “normal” samples.

We applied a two-way ANOVA to identify differentially expressed genes between normal and tumor tissue samples. The log_2_-transformed TPM values for Gene and Tissue were included as fixed factors, along with one interaction term to investigate gene-level effects for normal and tumor tissues (Gene by Tissue interaction term). For each gene, we conducted a comparison between normal and tumor samples blocked by the Gene factor and then determined significance by adjusting the *p*-value using the false discovery rate algorithm provided for in the R-package (BH corrected for normal versus tumor expression) calculations performed in R using multcomp_1.4-17 and emmeans_1.7.0 packages ran in R version 4.1.2 (1 November 2021) with RStudio front end (RStudio 2021.09.0+351 “Ghost Orchid” Release). Bar chart graphics were constructed using the ggplot2_3.3.5 R package.

We used a two-way hierarchical clustering technique to organize expression patterns such that samples and Gene expression displaying similar mRNA expression profiles were grouped using the average distance metric (default Euclidean distance implemented using the heatmap.2 function in the R package gplots_3.1.1). The cluster figure displayed mean expression levels in tumor tissue relative to normal ovarian tissue, with fold increases log_2_-transformed. The associated dendrograms organized and depicted expression levels of co-regulated genes for both (rows) and ovarian cancer patients (columns).

### 4.4. OS Analysis Using Kaplan–Meier Comparisons Verified for Taxol-Treated Patients in an Independent KMplotter Dataset for TGFB2-Independent Biomarkers Discovered in the TCGA Dataset

The KMplotter web tool (https://www.kmplot.com/analysis/index.php?p=service&cancer=ovar, accessed on 4 February 2024) was utilized to access ovarian cancer data from the Affymetrix dataset (*n* = 793 patients treated with taxol) for the selection of genes identified from the TCGA dataset (by cross referencing the KMplotter genes (HR > 1 and *Adj. p* < 0.05) from downloading Appendix A “11357_2023_742_MOESM3_ESM.xlsx” from the link https://static-content.springer.com/esm/art%3A10.1007%2Fs11357-023-00742-4/MediaObjects/11357_2023_742_MOESM3_ESM.xlsx (accessed on 23 April 2024) obtained from https://link.springer.com/article/10.1007/s11357-023-00742-4#Sec16, accessed on 23 April 2024) [31,32]. Patients were grouped based on the median cutoff for gene expression levels to compare the impact of high- and low-expression patient subgroups (Follow-up threshold = 120 months). The web tool reported the hazard ratio (HR) and log-rank *p*-values [31,32,129].

Patient-level treatment information was not supplied in the KMplotter database, but the portal enabled comparisons of sub-groups of treatments containing Taxol using Kaplan–Meier analysis. In these comparisons, we compared high versus low expression of prognostic marker genes (median cut offs) for patients with optimal debulking (*n* = 413) and patients with suboptimal debulking (*n* = 260). We also compared high versus low expression of prognostic marker genes for patients diagnosed with stage 1 + 2 (*n* = 69) and stage 3 + 4 cancers (*n* = 720).

### 4.5. Identifying Prognostically Relevant Proteins Associated with TGFB2-Dependent and TGFB2-Independent Marker Genes Using the STRING Interaction Algorithm

PPI networks were analyzed for ovarian cancer biomarkers that are markedly upregulated and demonstrated either *TGFB2* mRNA-dependent or -independent prognostic significance. Utilizing the STRING database, the objective was to delineate clusters of associations among these prognostic markers and to retrieve supporting literature regarding their relevance in cancers. The networks were built with STRING version 12 (https://string-db.org/, accessed on 27 October 2025) to identify possible hub proteins linking marker gene expression to protein interactions in the network [130]. In these diagrams, nodes represent protein identifiers, and edges represent associations between proteins (see Appendix A).

## 5. Conclusions

For *TGFB2*-dependent biomarkers, elevated *TGFB2* mRNA expression is a prognostic biomarker associated with enhanced overall survival (OS) outcomes in combination with low levels of expression of three gene targets: *TRPV4*, *STAU2*, and *HOXC4*; whose high levels of expression were correlated with worse survival outcomes. Therefore, serous ovarian cancer patients with high *TGFB2* expression and low expression of associated markers showed improved OS. A multivariate analysis revealed that, after controlling for *TGFB2* expression, four genes (*CLIC3*, *ANPEP*/*LAP1*, RIN2, and EMP1) showed negative correlations between mRNA expression and OS across all expression levels in an ovarian cancer cohort. These findings were validated using an independent KMplotter dataset focused on patients with Taxol-treated ovarian cancer. Consequently, this study identifies *TGFB2* as a significant biomarker among patients with high expression. It proposes an additional set of four *TGFB2*-independent prognostic markers to guide treatment for patients with Taxol-treated ovarian cancer.

## Figures and Tables

**Figure 1 ijms-26-11900-f001:**
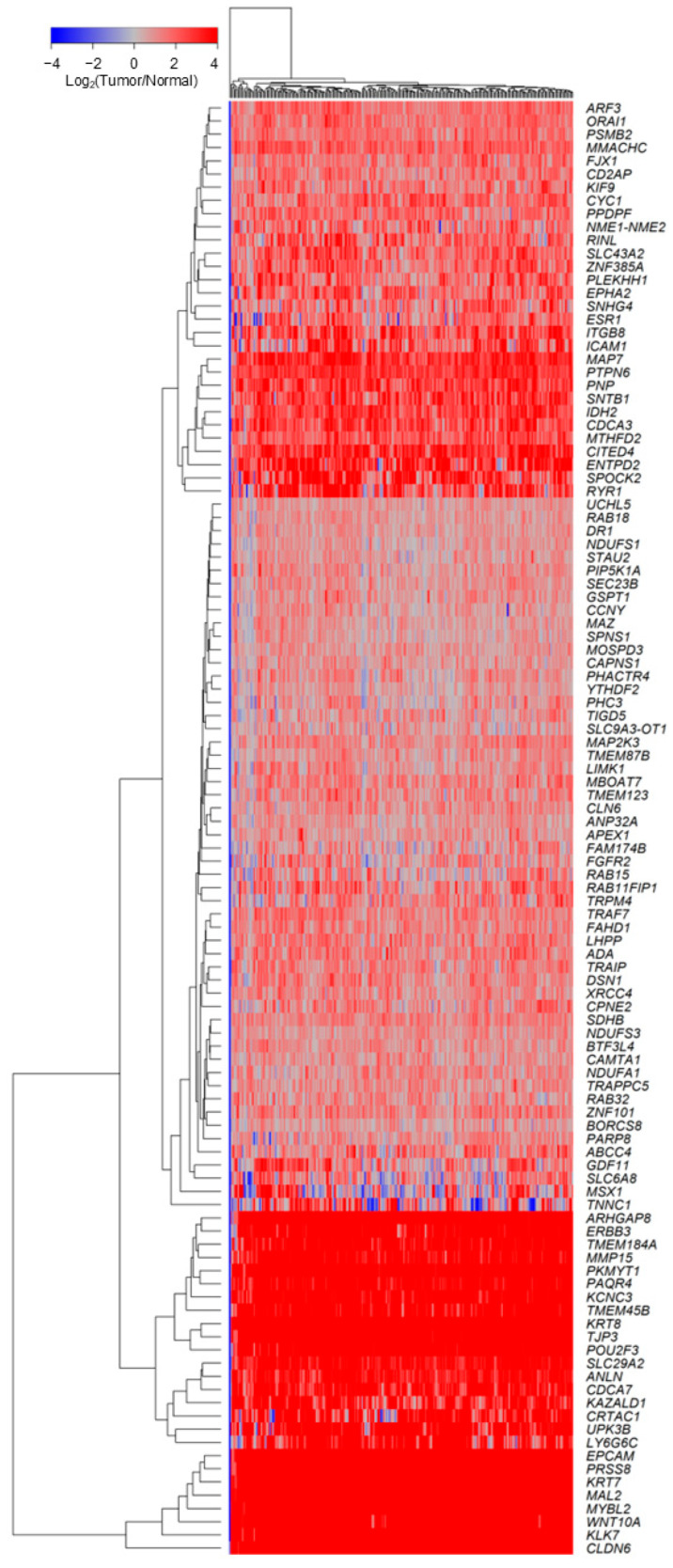
*TGFB2* mRNA-dependent prognostic markers identified by multivariate Cox proportional hazards modeling were significantly upregulated in tumor tissues. Multivariate analyses used the Cox proportional hazards model to assess the individual effects of *TGFB2*, Gene2 marker mRNA expression levels, and the *TGFB2*-dependent OS impact via the interaction term in the multivariate model (Model 1). This analysis was controlled for age at diagnosis (Mean age = 59 year) and treatment (chemo-only versus all other treatments). The expression of these genes is shown in the cluster figure, which uses a two-way hierarchical clustering technique to visualize co-regulated gene sets.

**Figure 2 ijms-26-11900-f002:**
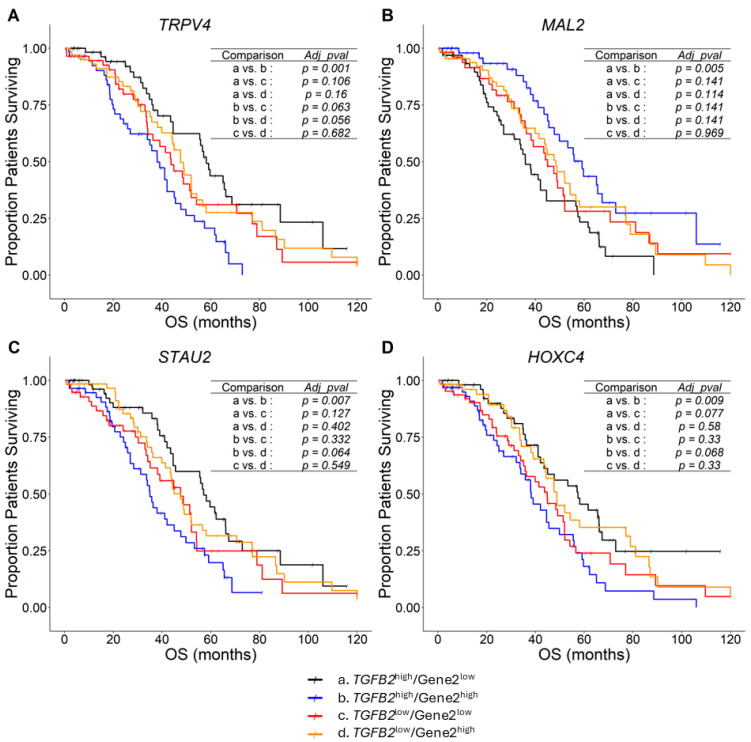
Prognostic markers for Ovarian cancer dependent on *TGFB2* mRNA expression assessed using Kaplan–Meier plots. Four genes were upregulated in tumors (*TRPV4*: 10-fold increase (*Adj. p* < 0.0001); *MAL2*: 705-fold increase (*Adj. p* < 0.0001); *STAU2*: 1.7-fold increase (*Adj. p* = 0.0002); and, *HOXC4*: 1.3-fold increase (*Adj. p* = 0.047)) and exhibited significant prognostic impact of the marker gene at high levels of *TGFB2* mRNA expression. Ovarian patients were correlated to OS outcomes investigating the impact of mRNA expression of *TGFB2* (median cut-off for high and low levels of expression), further stratified into four groups based on gene expression levels of *TRPV4* (**A**); *MAL2* (**B**); *STAU2* (**C**); and, *HOXC4* (**D**) mRNA expression levels (median cut-off for high and low levels) in these patients. The Kaplan–Meier plots show four stratified curves for each marker gene. Six pairwise comparisons were made between the four patient groups (*p*-values adjusted using the BH correction shown in the table insets). We identified *TGFB2*/Gene2 combinations that exhibited significant separation of the two arms of the four survival curves (*Adj. p* < 0.01).

**Figure 3 ijms-26-11900-f003:**
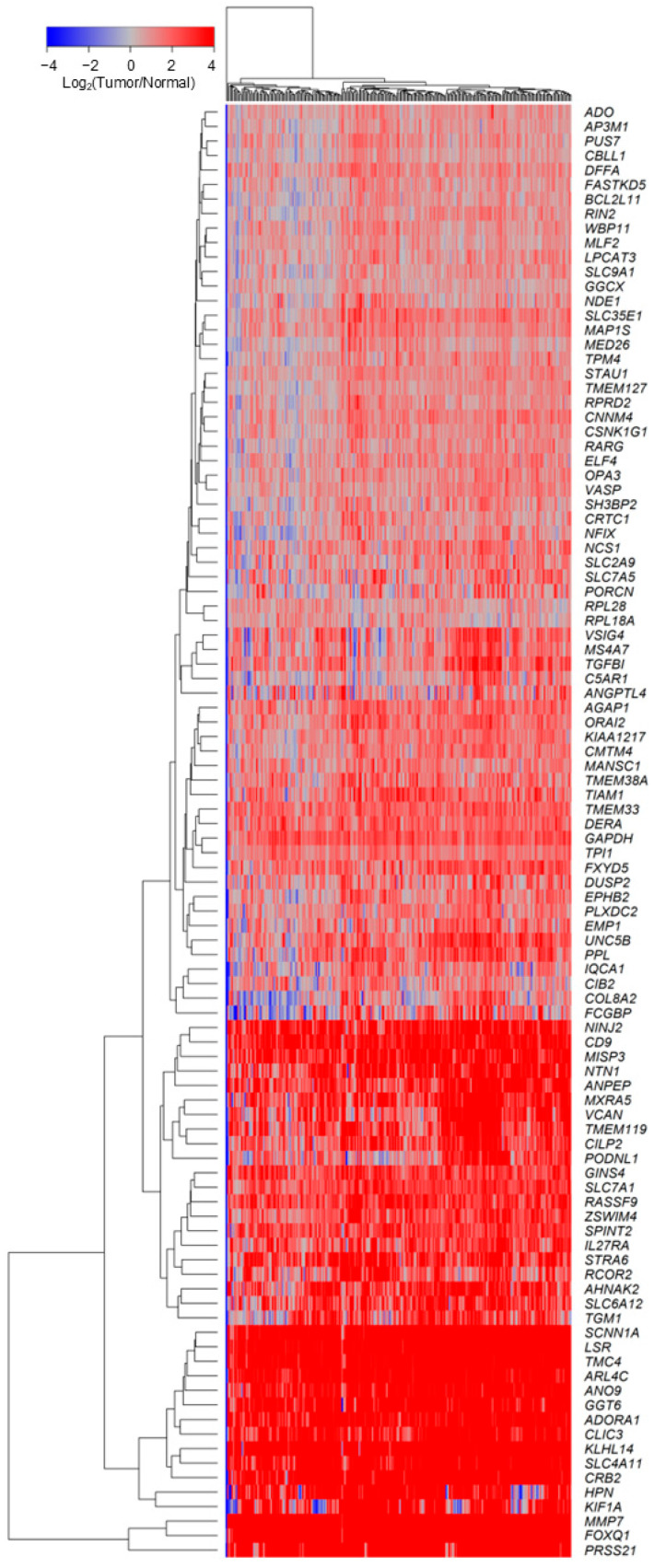
Negative prognostic markers identified from the multivariate Cox proportional hazards model were upregulated in tumor tissues. Multivariate analyses utilized the Cox proportional hazards model to assess the individual effects of *TGFB2* and Gene2 marker mRNA expression levels. This analysis controlled for age at diagnosis, treatment (chemo-only versus all other treatments), and the interaction between *TGFB2* and Gene2. We compared the mRNA expression in tumor tissues relative to normal tissues (*n* = 88), of 841 Gene2 markers that showed a significant increase in HR for Gene 2 independent of *TGFB2* mRNA expression resulting in 100 significantly upregulated in tumor tissues (*Adj. p* < 0.001, fold increase > 1.5, log_2_ (TPM) Expression in tumor tissue > 2) for 239 evaluable patients.

**Figure 4 ijms-26-11900-f004:**
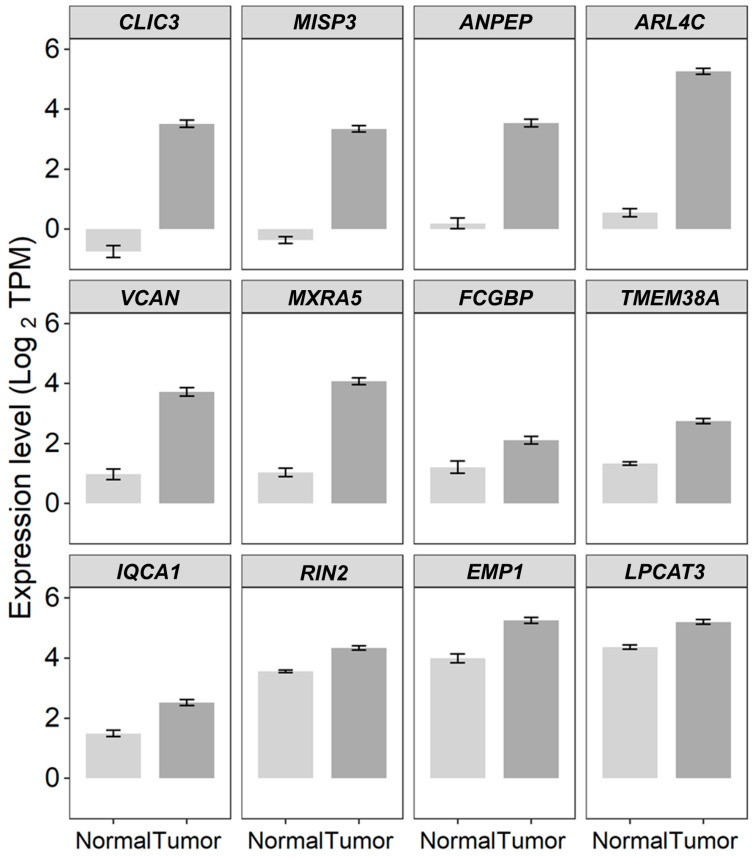
Gene expression levels have a significant prognostic impact, as shown in both TCGA and KMplotter datasets. The bar charts display the expression values (Mean ± SEM) of 12 significantly upregulated genes in ovarian cancer tumor tissues from the TCGA dataset, which were cross-referenced in KMplotter datasets for all patients.

**Figure 5 ijms-26-11900-f005:**
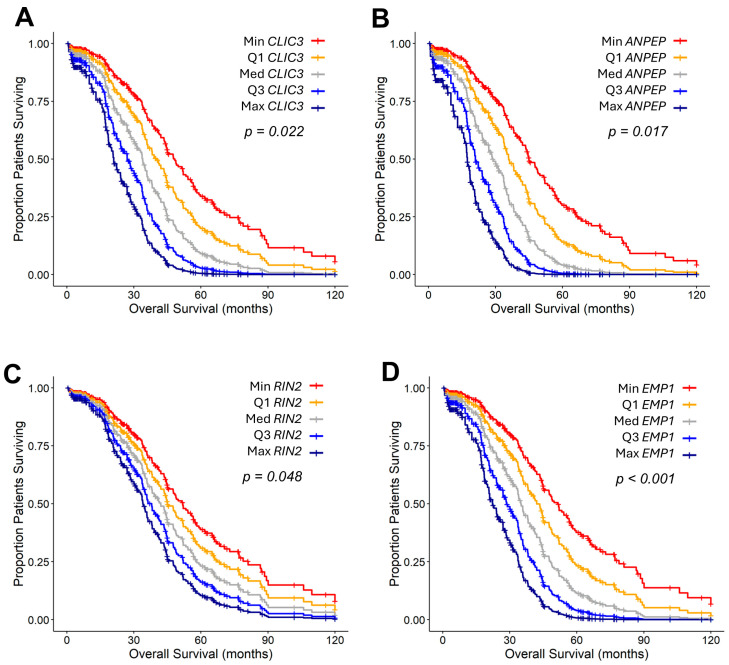
Predicted negative prognostic impacts were evaluated using the TCGA cohort, cross-referenced with the KMplotter database for Taxol-treated patients. OS curves generated from the multivariate Cox proportional hazards model (**A**–**D**) are shown for four genes (*CLIC3*, *ANPEP*/*LAP1*, *RIN2*, and *EMP1*). The shift in the baseline survival curves represented the predicted median OS times throughout the entire range of marker gene expression levels (Minimum (Min), Lower quartile (Q1), Median (Med, Upper Quartile (Q3), and Maximum (Max)), and the magnitude of the hazard ratio calculation. These genes also showed a significant impact on OS (*Adj. p*-value < 0.05) using the KMplotter web tool to analyze ovarian cancer data from the Affymetrix dataset (*n* = 793 patients treated with Taxol), comparing high versus low expression using the median cut-off for the best-selected JetSet probe set.

**Figure 6 ijms-26-11900-f006:**
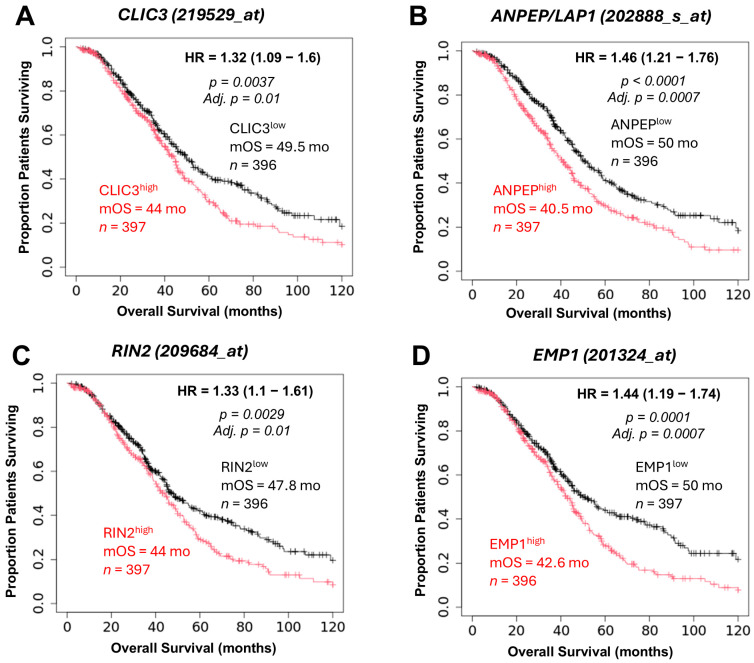
Validation of TCGA prognostic markers of predictive biomarkers using the independent KMplotter dataset for patients treated with Taxol. Four genes (*CLIC3*, *ANPEP/LAP1*, *RIN2*, and *EMP1*) significantly affected OS in both the TCGA and KMplotter datasets, as assessed using Kaplan–Meier plots. Using the KMplotter web tool to access ovarian cancer data from the Affymetrix dataset (*n* = 793 patients treated with taxol), high versus low expression was compared using the median cut-off for the best-selected JetSet probe set shown in the panels (**A**–**D**). The insets depict the univariate hazard ratio HR and 95% confidence limit calculations, and the mOS and numbers at risk at diagnosis (*n*). (**A**) Patients expressing high levels of *CLIC3* exhibited shorter median OS time (mOS = 44 months) compared to low levels of *CLIC3* expression (49.5 months). (**B**) Patients expressing high levels of *ANPEP*/*LAP1* exhibited shorter median OS time (40.5 months) compared to low levels of *ANPEP*/*LAP1* expression (50 months). (**C**) Patients expressing high levels of *RIN2* exhibited shorter median OS time (44 months) compared to low levels of *RIN2* expression (47.8 months). (**D**) Patients expressing high levels of *EMP1* exhibited shorter median OS time (42.6 months) compared to low levels of *EMP1* expression (50 months). The OS curves for *ANPEP*/*LAP1* shifted in parallel, whereas *CLIC3*, *RIN2*, and *EMP1* diverged beyond the mOS time. Red and black curves indicate high and low levels of Gene2 expression, respectively.

**Table 1 ijms-26-11900-t001:** Summary HR and fold increase for the prognostic markers.

Gene 2	Gene 2 mRNA (Z Score)		*TGFB2* by Gene 2 Interaction		Fold Increase (95% CI)	Contrast
	HR (95% CI)	*p*-Value	HR (95% CI)	*p*-Value		*Adj. p*-Value
*HOXC4*	1.36 (1.03–1.78)	0.028	1.74 (1.19–2.56)	0.004	1.31 (1.13–1.52)	0.047
*STAU2*	0.99 (0.87–1.13)	0.925	1.34 (1.07–1.67)	0.011	1.66 (1.51–1.82)	0.00018
*TRPV4*	1.26 (1.03–1.55)	0.027	1.53 (1.1–2.14)	0.012	9.96 (9.08–10.98)	<0.0000001
*MAL2*	1 (0.82–1.23)	0.974	0.79 (0.62–1)	0.046	705 (614.8–806.8)	<0.0000001
*EMP1*	1.32 (1.12–1.56)	<0.001	1.26 (0.93–1.7)	0.136	2.39 (2.09–2.74)	<0.0000001
*ANPEP*	1.45 (1.07–1.98)	0.017	1.79 (0.96–3.32)	0.066	10.16 (8.55–12.17)	<0.0000001
*CLIC3*	1.22 (1.03–1.45)	0.022	0.95 (0.74–1.23)	0.715	19.25 (16.28–22.55)	<0.0000001
*RIN2*	1.17 (1–1.36)	0.048	1.02 (0.84–1.24)	0.869	1.71 (1.53–1.90)	0.000024

## Data Availability

We analyzed clinical metadata (https://www.cbioportal.org/study/summary?id=ov_tcga_pan_can_atlas_2018; accessed on 28 November 2023) and RNA sequencing data (“data_mrna_seq_v2_rsem.txt”: Batch normalized from Illumina HiSeq_RNASeqV2 using the RSEM algorithm (Transcripts per million (TPM)) from 241 patients diagnosed with Ovarian cancer to compare OS correlates to mRNA expression values. The KMplotter web tool (https://www.kmplot.com/analysis/index.php?p=service&cancer=ovar, accessed on 4 February 2024) was utilized to access ovarian cancer data from the Affymetrix dataset (*n* = 793 patients treated with Taxol).

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
