# Peer review of "Bioinformatic Approach to Identify Potential *TGFB2*-Dependent and Independent Prognostic Biomarkers for Ovarian Cancers Treated with Taxol"

_ijms, 2025, doi:10.3390/ijms262411900_

Round 1

Reviewer 1 Report

Comments and Suggestions for Authors

The manuscript is too long, with a very detailed abstract and introduction and lengthy narrative sections throughout. The extensive background information makes the paper hard to follow and diminishes the impact of the main scientific message. I suggest significantly shortening and simplifying, especially in the abstract and introduction, to emphasize the key findings and their significance while removing redundant or overly complex descriptions.

STRING and PPI network analyses are descriptive and lack functional enrichment, pathway clustering, or mechanistic interpretation aligned with Taxol response. The PPI networks seem added rather than integral to the biomarker discovery process.

The manuscript still contains a lot of duplicated and repetitive content, indicating it needs major rewriting to reduce similarity and ensure originality.

SSome findings, particularly those with extreme fold changes and high hazard ratios, seem biologically improbable and may result from noise or inadequate filtering. Several conclusions are made beyond the supporting evidence.

Author Response

The manuscript is too long, with a very detailed abstract and introduction and lengthy narrative sections throughout. The extensive background information makes the paper hard to follow and diminishes the impact of the main scientific message.

  1. I suggest significantly shortening and simplifying, especially in the abstract and introduction, to emphasize the key findings and their significance while removing redundant or overly complex descriptions.

We thank the reviewer for the suggestions to simplify the manuscript. We have now revised the abstract and introduction sections. We have also shortened some of the descriptions in the discussion section.

“High-grade serous ovarian carcinoma is the most common and aggressive form of ovarian cancer, accounting for over 60% of cases and nearly 75% of deaths, mainly due to late diagnosis and tumor aggressiveness. Standard treatment is platinum-based chemotherapy with paclitaxel, but relapse is frequent. This study aimed to identify prognostic biomarkers for patients with poor survival outcomes after Taxol treatment using bioinformatics analysis. We examined the effects of TGFB2 mRNA expression and other markers on overall survival in serous ovarian cancer using the TCGA database, applying a multivariate Cox model that included interaction terms to identify TGFB2-dependent and independent prognostic markers, and controlling for age and treatment type. Candidate TGFB2-independent prognostic markers from TCGA were further validated using patient data from the KMplotter database. High TGFB2 mRNA expression emerged as a prognostic biomarker for three potential gene targets (TRPV4, STAU2, and HOXC4) associated with improved OS at low levels of gene target expression, we identified four additional markers (CLIC3, ANPEP/LAP1, RIN2, and EMP1) that exhibited a TGFB2-independent negative correlation between mRNA expression and OS across the full spectrum of gene expression values in the ovarian cancer cohort validated using independent dataset from KMplotter, for Taxol-treated ovarian cancer patients. This study proposes a panel of potential prognostic biomarkers for the treatment of ovarian cancer patients, particularly by leveraging TGFB2-dependent mRNA expression as a significant biomarker, alongside four additional TGFB2-independent prognostic markers, for patients undergoing Taxol-based therapies. Future prospective clinical trials will be required to validate these prognostic markers.” (Lines 13-34)

  1. STRING and PPI network analyses are descriptive and lack functional enrichment, pathway clustering, or mechanistic interpretation aligned with Taxol response. The PPI networks seem added rather than integral to the biomarker discovery process.

On reflection, we agree with the reviewer’s comment.  The PPI analysis is primarily a discussion point rather than an integral component of biomarker selection. Accordingly, we have moved the figures to Supplementary Figures S1 and S2.  We have revised the methods section to clarify the intention of the PPI analysis.

“PPI networks were analyzed for ovarian cancer biomarkers that are markedly upregulated and demonstrated either TGFB2 mRNA-dependent or -independent prognostic significance. Utilizing the STRING database, the objective was to delineate clusters of associations among these prognostic markers and to retrieve supporting literature regarding their relevance in cancers. The networks were built with STRING version 12 (https://string-db.org/, accessed 27th October 2025) to identify possible hub proteins linking marker gene expression to protein interactions in the network [130]. In these diagrams, nodes represent protein identifiers, and edges represent associations between proteins (see Supplementary Methods). ” (Lines 800-808).

  1. The manuscript still contains a lot of duplicated and repetitive content, indicating it needs major rewriting to reduce similarity and ensure originality.

In accordance with this suggestion, we have removed duplicate descriptions of results in the discussion section and attempted to reduce the amount of explanatory information.

  1. SSome findings, particularly those with extreme fold changes and high hazard ratios, seem biologically improbable and may result from noise or inadequate filtering.

We have included a flowchart in the supplementary section to illustrate the filtering and hypothesis-testing steps in the analytical procedure.  The filtering primarily occurred during parameter selection for the multivariate Cox proportional hazards model.  Comparisons between normal and tumor samples were performed to identify differentially expressed genes in the tumor.  At this stage, we ensured that the signal was detectable in the tumor (Log2(TPM) > 2).  The large fold-increases were observed due to very low expression in normal tissues.

“Prefiltering was performed by selecting genes with a mean TPM greater than 10 across all ovarian cancer patient samples for further analysis. Potential prognostic marker genes were identified using LRTs, which were used to filter and preselect genes before conducting further statistical analyses, including comparisons of normal versus tumor expression and Kaplan-Meier survival analyses. Specifically, the LRT compared two nested models: a null model, in which all coefficients were set to zero, and the dependent variables were assumed not to affect the HR, and an alternative model that included all dependent variables in the regression. The maximized log-likelihoods from both models were used to compute p-values, thereby assessing whether including the dependent variables significantly improved model fit. For downstream analysis, we retained marker genes with p-values less than 0.05. This filtering approach increased the statistical power of subsequent formal tests by narrowing the number of hypotheses considered before multiple-testing adjustments of p-values for differential expression in normal versus tumor comparisons.” (Lines 707-720)

  1. Several conclusions are made beyond the supporting evidence.

We have revised the limitations paragraph to reflect that this study was designed to generate hypotheses for subsequent testing and validation.

“The KMplotter dataset did not provide patient-level treatment information thereby presenting our study with a limitation of insufficient treatment regimen matching, that will require future head-to-head clinical trials for validation.  This study is in the bioinformatics screening stage, generating hypotheses for further testing; due to limited access to clinical samples, Reverse Transcription Polymerase Chain Reaction (qRT-PCR) / Immunohistochemistry (IHC) validation has not been conducted. In future work, we will collect clinical samples from paclitaxel-treated ovarian cancer patients to validate protein expression levels of the biomarkers and the consistency between mRNA and protein expression. The TGFB2-dependency will require additional measurements of mRNA levels of TGFB2 along with the prognostic markers in the cellular compartments of the tumor. Thus, a validation pipeline combining qRT-PCR, IHC, and single-cell RNA-seq studies will enable comprehensive confirmation of biomarkers by integrating bulk quantitative mRNA measurement, spatial protein localization, and single-cell resolution of tumor heterogeneity. Additionally, preclinical studies will be required to evaluate the bioequivalence of the pharmacokinetic and efficacy profiles of LNP-formulated versus unformulated Taxol in patients with high levels of TGFB2 and the companion markers, as well as the TGFB2-independent prognostic markers.” (Lines 641-657)

We have also scaled back some of the claims eg.

“TRPV4 is therefore a potential therapeutic target for ovarian cancer [47]. Additional research is required to confirm this target through protein-level analyses.” (Line 364)

Our results suggest that patients with high MAL2 and TGFB2 expression have improved survival and warrant further validation through protein-level analysis.” (Line 377)

Reviewer 2 Report

Comments and Suggestions for Authors

Summary
Brief bioinformatics study asking whether TGFB2‐dependent or TGFB2‐independent mRNA markers predict overall survival in serous ovarian cancer, with a focus on Taxol-treated patients. The authors screen ~16k genes in TCGA using multivariable Cox models that include TGFB2, a Gene2 term, their interaction, age at diagnosis, and a coarse “chemo-only vs other” treatment flag; tumor-vs-normal expression is contrasted using GTEx normals (n=88). They report four TGFB2-dependent candidates (TRPV4, MAL2, STAU2, HOXC4) and four TGFB2-independent markers (CLIC3, ANPEP/LAP1, RIN2, EMP1) that validate by KM plotter in a Taxol-treated microarray cohort (N=793; median cutoffs). Overall, it’s a clear hypothesis-generating analysis, but several modeling choices and claims need tightening before the biomarkers feel actionable.

Strengths

Clear framing (TGFB2 as a context variable) and a prespecified multivariable Cox setup with an interaction term.

Cross-dataset check: TCGA discovery → KM plotter limited to Taxol-treated cases

Markers are shown both as tumor-vs-normal upregulated and as prognostic, which helps biological plausibility.

Recommendations

In TCGA, “chemo-only vs other” is not the same as known paclitaxel exposure. Please show, if possible, a paclitaxel-specific indicator (or admit it’s unavailable) and re-estimate the Cox models, or clearly state that Taxol specificity comes only from the KMplotter step.

Control FDR at the gene-screening stage and again for the KM validations (BH or Storey). Avoid median dichotomization where possible

Add stage, debulking status/residual disease, BRCA/HRD, and bevacizumab/PARP exposure if available

Treat STRING clusters and the AI-assisted abstract mining as supportive only; they shouldn’t drive conclusions. Move tool lists and UI details to Supplement and provide the specific STRING parameters/files for reproducibility.

For each of the eight highlighted genes, provide: tumor/normal fold-change with CI, adjusted HR per SD (continuous) and KM curves shown only as illustration

Conclusion
The TGFB2-context idea is sensible, and the short list (TRPV4, MAL2, STAU2, HOXC4; CLIC3, ANPEP, RIN2, EMP1) is a reasonable starting point. Firm up multiple-testing control, add key clinical covariates, and be precise about Taxol exposure. With stronger statistics, cleaner treatment annotation, and one more independent check, this would read as a solid, hypothesis-generating biomarker paper rather than a preliminary screen.

Author Response

Comment 1

  1. In TCGA, “chemo-only vs other” is not the same as known paclitaxel exposure. Please show, if possible, a paclitaxel-specific indicator (or admit it’s unavailable) and re-estimate the Cox models, or clearly state that Taxol specificity comes only from the KMplotter step.

We thank the reviewer for these constructive comments.  We have now expanded the methods section to provide a full description of the treatment regimens. In the TCGA data set 170 out of the 191 (90%) chemo-only patients were treated with Taxol, thereby aligning the cohort with the KMplotter cohort.

“Patient-level treatment information (treatment regimen reported as “TREATMENT_TYPE”, “AGENT”, “START_DATE”, “STOP_DATE”, “NUMBER_OF_CYCLES”) was available for the ovarian cancer dataset (n = 241) that was utilized to stratify patients based on exposure to chemotherapy (n = 238; with most common treatment regimens containing carboplatin (n = 219), paclitaxel (n = 215), doxorubicin (n = 80), cisplatin (n = 72), and topotecan (n = 68)); targeted molecular therapy (most common being bevacizumab (n = 29); radiation therapy (n = 17); hormone therapy (most common being Tamoxifen (n = 17)); immunotherapy (n = 5). The chemotherapy-treated group of patients was further subdivided into two groups: chemo-only for patients that were not exposed to targeted molecular therapy, radiation therapy, hormone therapy, or immunotherapy (n = 191); and others (n = 50). We also determined the number of patients that were exposed to paclitaxel from the chemo-only group (Taxol; n = 170), patients exposed to chemotherapy but not paclitaxel or other treatment regimens (other chemo-only; n = 20), and others (n = 50).” (Lines 679-691)

We also performed an additional Cox proportional hazards model (Supplementary Figure 2; Model 2)  that showed overlapping 95% CI for 170 Taxol treated patients and patients treated with other chemotherapies (n = 20).  The increases in HR were maintained for all 4 prognostic markers.

Comment 2

  1. Control FDR at the gene-screening stage and again for the KM validations (BH or Storey). Avoid median dichotomization where possible

We have now provided a flow chart in the supplementary methods section that indicates the pre-filtering and hypothesis testing steps with BH-adjusted p-values, and we have indicated the comparisons with BH corrections in the main manuscript.

Comment 3

  1. Add stage, debulking status/residual disease, BRCA/HRD, and bevacizumab/PARP exposure if available

The KMplotter dataset made available stage and debulking status, and thus we have included new supplementary analyses showing prognostic impact for optimal debulking and stage3 + 4 stratified patients (Supplementary figures 4 and 5). Bevacizumab treatment information was provided for the TCGA dataset allowing us to re-examine the impact of this targeted therapy in the Cox regression Model 3 (Supplementary figure 3).

We expanded the results section to include:

“The availability of patient-level treatment metadata for the TCGA data set enabled us to investigate the impact of chemo-only further (Figure S3; Model 1), exposure to Taxol (Figure S4; Model 2), and patients receiving bevacizumab therapy (Figure S5; Model 3) using the multivariate Cox proportional hazards model.  Examination of the Akaike information criterion (AIC) and LRT p-values showed no appreciable improvement in the fit of the additional models 2 and 3 to the data (Figures S3-S5; increases in AIC and LRT p-values were marginal and showed increases comparing Model 1 with Model 2, and comparing Model 1 with Model 3 (except for RIN2)). Model 1 provided the best fit to the data, and 171 of 191 (90%) patients in the chemo-only group were treated with Taxol (Figure S3). We then re-examined the Taxol-treated group (n = 171), patients treated with other chemotherapy agents (n = 20), and patients treated with other treatment modalities (n = 50) (Figure S4; Model 2).  The application of Model 2 maintained the HR increases observed for all four prognostic markers (p < 0.05). Patients exposed to Taxol (n = 171) exhibited increases in HR that did not achieve statistical significance (HR ranged from 1.39 to 1.41). The cohort of patients exposed to bevacizumab (Figure S5; n = 29) showed improved survival (decreased HR), but the difference was not statistically significant.

The KMplotter data portal facilitated the investigation of patients' stage and debulking status (Figures S6 and S7). In this analysis, patients presenting with stages 3 or 4 cancers (n = 720) showed significantly worse OS curves at high mRNA levels (median cut-offs) for all four prognostic markers (Figure S6; HR ranged from 1.27 to 1.51). Separation of OS curves was also observed for all four prognostic markers in patients undergoing optimal debulking (Figure S7; HRs ranged from 1.33 to 1.84), comparing patients with high versus low levels of the prognostic markers.” (Lines 285-307)

Comment 4

  1. Treat STRING clusters and the AI-assisted abstract mining as supportive only; they shouldn’t drive conclusions. Move tool lists and UI details to Supplement and provide the specific STRING parameters/files for reproducibility.

We have now moved these sections to the supplementary methods and provided the STRING parameters in the description

Comment 5

  1. For each of the eight highlighted genes, provide: tumor/normal fold-change with CI, adjusted HR per SD (continuous) and KM curves shown only as illustration

We have now included summary table 1 in the main manuscript as suggested.

Conclusion

The TGFB2-context idea is sensible, and the short list (TRPV4, MAL2, STAU2, HOXC4; CLIC3, ANPEP, RIN2, EMP1) is a reasonable starting point. Firm up multiple-testing control, add key clinical covariates, and be precise about Taxol exposure. With stronger statistics, cleaner treatment annotation, and one more independent check, this would read as a solid, hypothesis-generating biomarker paper rather than a preliminary screen.

Reviewer 3 Report

Comments and Suggestions for Authors

This study focuses on screening prognostic biomarkers for paclitaxel (Taxol)-treated high-grade serous ovarian carcinoma (HGSOC). By integrating data from The Cancer Genome Atlas (TCGA) and KMplotter databases using bioinformatics approaches, it explores TGFB2-dependent and TGFB2-independent prognostic genes. The research direction aligns with clinical needs, with a clear technical route, and its core findings hold potential for clinical translation. However, the manuscript has obvious deficiencies in format standardization, data description, methodological logic, and interpretation of limitations, requiring targeted revisions and improvements. It is recommended for acceptance after minor revisions. Specific Issues and Revision Suggestions (I) Issues with Gene Symbol Format and Terminology Standardization Problem The manuscript fails to follow the italicization rules for gene and protein symbols: all gene symbols (e.g., TGFB2, TRPV4, CLIC3) are not italicized, and no clear formatting distinction is made between gene and protein symbols. Additionally, when "TGFB2" is first mentioned, its full name "Transforming Growth Factor Beta 2 Gene" is not provided, and the formatting of "biomarkers" is ambiguous when referring to either genes or proteins. Revision Suggestions: Standardize formatting throughout the manuscript: Use italics for gene symbols (e.g., TGFB2, CLIC3) and upright font for protein symbols (e.g., TGF-β2, CLIC3 protein). Add the full name of "TGFB2" when it first appears in the text: "Transforming Growth Factor Beta 2 Gene (TGFB2)". (II) Typographical Errors in Tissue Type Description Problem The research focuses on ovarian cancer, yet Line 870 mentions "normal pancreas expression levels" and "pancreatic cancer patients", which conflict with the study theme and are confirmed to be typographical errors. In contrast, the description in Line 118 regarding "the role of TGFB2 in pancreatic ductal adenocarcinoma (PDAC)" is a reasonable citation of background research and does not conflict with the current study. Revision Suggestions: Correct Line 870 by changing "normal pancreas" to "normal ovary" and "pancreatic cancer patients" to "ovarian cancer patients". Conduct a full-text search for the terms "pancreas/pancreatic" and correct all descriptions that conflict with the ovarian cancer research theme, except for citations related to PDAC background. (III) Inconsistency Between Figure Labeling and Quantity Problem Figure 6 contains four subfigures (A-D) corresponding to four genes, but the first sentence of the figure legend states "OS curves generated from the multivariate Cox proportional hazards model (A-C)". The notation "(A-C)" contradicts the number of subfigures and genes, which is confirmed to be a typographical error. Revision Suggestion: Revise "(A-C)" to "(A-D)" in the legend of Figure 6. Additionally, check the consistency between "subfigure labels" and "the number of items mentioned in the main text" for all figures and tables throughout the manuscript. (IV) Errors in Methodological Description and Ambiguity in Variable Definition Problem In Line 891, "v0." should be "(v)", resulting in inconsistent numbering format. Furthermore, the operational definition of "treatment regimen (Chemo-only vs. other treatments)" is unclear—for example, whether "Chemo-only" includes paclitaxel, and whether "other treatments" include targeted therapy—leading to ambiguous variable definition. Revision Suggestions: Correct the symbol error by changing "v0." to "(v)" and standardize the numbering format to "(i)(ii)(iii)(iv)(v)". Add the definition of "treatment regimen" in the "Materials and Methods" section: "‘Chemo-only’ refers to patients who received chemotherapy alone (including platinum-paclitaxel combination therapy and paclitaxel monotherapy); ‘other treatments’ refer to combined regimens such as chemotherapy plus targeted therapy or surgery plus chemotherapy." Cite the original annotation of "treatment regimen" from the TCGA database to support this definition. (V) Issues with Treatment Regimen Matching and Biomarker Specificity Problem Biomarkers were initially screened in TCGA using "Chemo-only" patients (without restricting to paclitaxel-containing regimens), then validated in KMplotter using "paclitaxel-treated" patients. The proportion of TCGA "Chemo-only" patients who received paclitaxel is not specified, leading to potential confounding risks from inconsistent treatment regimens. Additionally, no comparison was made of hazard ratio (HR) differences of biomarkers between the "paclitaxel-treated group" and "non-paclitaxel-treated group", making it impossible to confirm paclitaxel specificity. Revision Suggestions: Add a "Table of Treatment Regimen Distribution in Patients from the Two Databases" to clarify the proportion of TCGA "Chemo-only" patients who received paclitaxel and quantify the matching degree of treatment regimens. Supplement an interaction term analysis of "treatment regimen × biomarker expression" to compare HR differences of biomarkers between the paclitaxel group and non-paclitaxel group, thereby verifying specificity. Discuss the "limitation of insufficient treatment regimen matching" in the "Discussion" section and explain that subsequent head-to-head clinical trials are needed for further validation. (VI) Other Supplementary Improvement Issues Lack of Patient Baseline CharacteristicsNo comparison was made of baseline characteristics (e.g., age, tumor stage, BRCA mutation status) between TCGA and KMplotter patients, which may introduce sampling bias. It is recommended to add a "Table of Baseline Characteristics Comparison" and conduct a stratified analysis to assess the impact of baseline factors on the prognostic effect of biomarkers. Insufficient Description of Laboratory ValidationThe manuscript only mentions that qRT-PCR/IHC validation was not performed, without explaining the reasons (e.g., sample accessibility limitations) or discussing the impact of potential discrepancies between mRNA and protein expression. It is recommended to add: "This study is in the bioinformatics screening stage; due to limited access to clinical samples, qRT-PCR/IHC validation has not been conducted. In future work, we will collect clinical samples from paclitaxel-treated ovarian cancer patients to validate protein expression levels of the biomarkers and the consistency between mRNA and protein expression." Unclear Screening Logic for Key GenesThe criteria for narrowing down 808 TGFB2-dependent genes to 4 key genes were not explained. It is recommended to add a "Flowchart of Key Biomarker Screening" to clarify the steps and thresholds of "differential expression analysis → survival association analysis → validation consistency". Lack of Comparison with Latest LiteratureMost references are from before 2024, and no studies on paclitaxel resistance biomarkers in ovarian cancer published in 2024–2025 were included. It is recommended to add a "Comparison with Latest Research" section in the "Discussion" to analyze the similarities and differences between the findings of this study and the latest advances.

Author Response

(I) Issues with Gene Symbol Format and Terminology Standardization Problem The manuscript fails to follow the italicization rules for gene and protein symbols: all gene symbols (e.g., TGFB2, TRPV4, CLIC3) are not italicized, and no clear formatting distinction is made between gene and protein symbols. Additionally, when "TGFB2" is first mentioned, its full name "Transforming Growth Factor Beta 2 Gene" is not provided, and the formatting of "biomarkers" is ambiguous when referring to either genes or proteins.

Revision Suggestions: Standardize formatting throughout the manuscript: Use italics for gene symbols (e.g., TGFB2, CLIC3) and upright font for protein symbols (e.g., TGF-β2, CLIC3 protein). Add the full name of "TGFB2" when it first appears in the text: "Transforming Growth Factor Beta 2 Gene (TGFB2)".

We have now fully implemented the reviewer’s suggestions.

(II) Typographical Errors in Tissue Type Description Problem The research focuses on ovarian cancer, yet Line 870 mentions "normal pancreas expression levels" and "pancreatic cancer patients", which conflict with the study theme and are confirmed to be typographical errors. In contrast, the description in Line 118 regarding "the role of TGFB2 in pancreatic ductal adenocarcinoma (PDAC)" is a reasonable citation of background research and does not conflict with the current study.

Revision Suggestions: Correct Line 870 by changing "normal pancreas" to "normal ovary" and "pancreatic cancer patients" to "ovarian cancer patients". Conduct a full-text search for the terms "pancreas/pancreatic" and correct all descriptions that conflict with the ovarian cancer research theme, except for citations related to PDAC background.

We have now fully implemented the reviewer’s suggestions.

(III) Inconsistency Between Figure Labeling and Quantity Problem Figure 6 contains four subfigures (A-D) corresponding to four genes, but the first sentence of the figure legend states "OS curves generated from the multivariate Cox proportional hazards model (A-C)". The notation "(A-C)" contradicts the number of subfigures and genes, which is confirmed to be a typographical error.

Revision Suggestion: Revise "(A-C)" to "(A-D)" in the legend of Figure 6. Additionally, check the consistency between "subfigure labels" and "the number of items mentioned in the main text" for all figures and tables throughout the manuscript.

We have now fully implemented the reviewer’s suggestions.

(IV) Errors in Methodological Description and Ambiguity in Variable Definition Problem In Line 891, "v0." should be "(v)", resulting in inconsistent numbering format. Furthermore, the operational definition of "treatment regimen (Chemo-only vs. other treatments)" is unclear—for example, whether "Chemo-only" includes paclitaxel, and whether "other treatments" include targeted therapy—leading to ambiguous variable definition.

Revision Suggestions: Correct the symbol error by changing "v0." to "(v)" and standardize the numbering format to "(i)(ii)(iii)(iv)(v)". Add the definition of "treatment regimen" in the "Materials and Methods" section: "‘Chemo-only’ refers to patients who received chemotherapy alone (including platinum-paclitaxel combination therapy and paclitaxel monotherapy); ‘other treatments’ refer to combined regimens such as chemotherapy plus targeted therapy or surgery plus chemotherapy."

Cite the original annotation of "treatment regimen" from the TCGA database to support this definition.

We have now fully implemented the reviewer’s suggestions. The manuscript now includes full description of the treatment regimens.

“Patient-level treatment information (treatment regimen reported as “TREATMENT_TYPE”, “AGENT”, “START_DATE”, “STOP_DATE”, “NUMBER_OF_CYCLES”) was available for the ovarian cancer dataset (n = 241) that was utilized to stratify patients based on exposure to chemotherapy (n = 238; with most common treatment regimens containing carboplatin (n = 219), paclitaxel (n = 215), doxorubicin (n = 80), cisplatin (n = 72), and topotecan (n = 68)); targeted molecular therapy (most common being bevacizumab (n = 29); radiation therapy (n = 17); hormone therapy (most common being Tamoxifen (n = 17)); immunotherapy (n = 5). The chemotherapy-treated group of patients was further subdivided into two groups: chemo-only for patients that were not exposed to targeted molecular therapy, radiation therapy, hormone therapy, or immunotherapy (n = 191); and others (n = 50). We also determined the number of patients that were exposed to paclitaxel from the chemo-only group (Taxol; n = 170), patients exposed to chemotherapy but not paclitaxel or other treatment regimens (other chemo-only; n = 20), and others (n = 50).” (Lines 678-691)

(V) Issues with Treatment Regimen Matching and Biomarker Specificity Problem Biomarkers were initially screened in TCGA using "Chemo-only" patients (without restricting to paclitaxel-containing regimens), then validated in KMplotter using "paclitaxel-treated" patients. The proportion of TCGA "Chemo-only" patients who received paclitaxel is not specified, leading to potential confounding risks from inconsistent treatment regimens.  Additionally, no comparison was made of hazard ratio (HR) differences of biomarkers between the "paclitaxel-treated group" and "non-paclitaxel-treated group", making it impossible to confirm paclitaxel specificity.

We also performed an additional Cox proportional hazards model (Supplementary Figure 2; Model 2)  that showed overlapping 95% CI for 170 Taxol treated patients and patients treated with other chemotherapies (n = 20).  The increases in HR were maintained for all 4 prognostic markers.

We expanded the results section to include:

“The availability of patient-level treatment metadata for the TCGA data set enabled us to investigate the impact of chemo-only further (Figure S3; Model 1), exposure to Taxol (Figure S4; Model 2), and patients receiving bevacizumab therapy (Figure S5; Model 3) using the multivariate Cox proportional hazards model.  Examination of the Akaike information criterion (AIC) and LRT p-values showed no appreciable improvement in the fit of the additional models 2 and 3 to the data (Figures S3-S5; increases in AIC and LRT p-values were marginal and showed increases comparing Model 1 with Model 2, and comparing Model 1 with Model 3 (except for RIN2)). Model 1 provided the best fit to the data, and 171 of 191 (90%) patients in the chemo-only group were treated with Taxol (Figure S3). We then re-examined the Taxol-treated group (n = 171), patients treated with other chemotherapy agents (n = 20), and patients treated with other treatment modalities (n = 50) (Figure S4; Model 2).  The application of Model 2 maintained the HR increases observed for all four prognostic markers (p < 0.05). Patients exposed to Taxol (n = 171) exhibited increases in HR that did not achieve statistical significance (HR ranged from 1.39 to 1.41). The cohort of patients exposed to bevacizumab (Figure S5; n = 29) showed improved survival (decreased HR), but the difference was not statistically significant.

The KMplotter data portal facilitated the investigation of patients' stage and debulking status (Figures S6 and S7). In this analysis, patients presenting with stages 3 or 4 cancers (n = 720) showed significantly worse OS curves at high mRNA levels (median cut-offs) for all four prognostic markers (Figure S6; HR ranged from 1.27 to 1.51). Separation of OS curves was also observed for all four prognostic markers in patients undergoing optimal debulking (Figure S7; HRs ranged from 1.33 to 1.84), comparing patients with high versus low levels of the prognostic markers.” (Lines 285-307)

Revision Suggestions: Add a "Table of Treatment Regimen Distribution in Patients from the Two Databases" to clarify the proportion of TCGA "Chemo-only" patients who received paclitaxel and quantify the matching degree of treatment regimens.

Patient-level treatment information was not provided for the KMplotter dataset; we could only subset the patients treated with Taxol-containing regimens.  The TCGA dataset provided the patient-level treatment information that enabled us to include two additional analyses (Supplementary figures 2 and 3).

Supplement an interaction term analysis of "treatment regimen × biomarker expression" to compare HR differences of biomarkers between the paclitaxel group and non-paclitaxel group, thereby verifying specificity.

We performed this analysis, and no significant interaction terms were found; the Cox proportional hazards model exhibited worse model fits and inflated errors for all the other parameters assessed by AIC and LRT-p values.  Therefore, we did not include these models in the manuscript. We instead examined the HR and 95% CI for Taxol and other chemo-treated cohorts in our Model 2 (Supplementary figure 2) parameters.

Discuss the "limitation of insufficient treatment regimen matching" in the "Discussion" section and explain that subsequent head-to-head clinical trials are needed for further validation.

We have expanded the discussion section to address regimen matching. We were not too concerned with Regimen matching, as 90% of the TCGA chemo-only patients were treated with Taxol.

“We conducted a cross-referential analysis of the 100 genes identified within the TCGA dataset (90% of the chemo-only cohort of patients were treated with Taxol, and evaluated using the multivariate Cox proportional hazards model) alongside the findings of Győrffy et al. (2023) [31,32], who established significant correlations OS and gene expression among all ovarian cancer patients, utilizing optimal cut-off values. To refine our candidate list of prognostic markers, we applied more stringent criteria, specifically focusing on median cut-off values to distinguish between high- and low-expressing Taxol-treated patient sub-groups.” (lines 507-514)

We did include the regimen matching limitation in the discussion section:

“The KMplotter dataset did not provide patient-level treatment information thereby presenting our study with a limitation of insufficient treatment regimen matching, that will require future head-to-head clinical trials for validation.” (Lines 646 to 648)

(VI) Other Supplementary Improvement Issues Lack of Patient Baseline CharacteristicsNo comparison was made of baseline characteristics (e.g., age, tumor stage, BRCA mutation status) between TCGA and KMplotter patients, which may introduce sampling bias. It is recommended to add a "Table of Baseline Characteristics Comparison" and conduct a stratified analysis to assess the impact of baseline factors on the prognostic effect of biomarkers.

The KMplotter dataset does not provide patient-level treatment information to facilitate direct comparison of baseline characteristics.  The KMplotter dataset made available stage and debulking status, and thus we have included new supplementary analyses showing prognostic impact for optimal debulking and stage3 + 4 stratified patients (Supplementary figures 4 and 5). Bevacizumab treatment information was provided for the TCGA dataset, allowing us to re-examine the impact of this targeted therapy in the Cox regression Model 3 (Supplementary figure 3).

Insufficient Description of Laboratory ValidationThe manuscript only mentions that qRT-PCR/IHC validation was not performed, without explaining the reasons (e.g., sample accessibility limitations) or discussing the impact of potential discrepancies between mRNA and protein expression. It is recommended to add: "This study is in the bioinformatics screening stage; due to limited access to clinical samples, qRT-PCR/IHC validation has not been conducted. In future work, we will collect clinical samples from paclitaxel-treated ovarian cancer patients to validate protein expression levels of the biomarkers and the consistency between mRNA and protein expression."

We thank the reviewer for this suggestion, and we have now included the following paragraph in the discussion:

“This study is in the bioinformatics screening stage, generating hypotheses for further testing; due to limited access to clinical samples, Reverse Transcription Polymerase Chain Reaction (qRT-PCR) / Immunohistochemistry (IHC) validation has not been conducted. In future work, we will collect clinical samples from paclitaxel-treated ovarian cancer patients to validate protein expression levels of the biomarkers and the consistency between mRNA and protein expression. The TGFB2-dependency will require additional measurements of mRNA levels of TGFB2 along with the prognostic markers in the cellular compartments of the tumor. Thus, a validation pipeline combining qRT-PCR, IHC, and single-cell RNA-seq studies will enable comprehensive confirmation of biomarkers by integrating bulk quantitative mRNA measurement, spatial protein localization, and single-cell resolution of tumor heterogeneity. Additionally, preclinical studies will be required to evaluate the bioequivalence of the pharmacokinetic and efficacy profiles of LNP-formulated versus unformulated Taxol in patients with high levels of TGFB2 and the companion markers, as well as the TGFB2-independent prognostic markers.” (Lines 648-662)

Unclear Screening Logic for Key GenesThe criteria for narrowing down 808 TGFB2-dependent genes to 4 key genes were not explained.

It is recommended to add a "Flowchart of Key Biomarker Screening" to clarify the steps and thresholds of "differential expression analysis → survival association analysis → validation consistency".

We thank the reviewer for this helpful suggestion and have now included the flowchart in the supplementary methods section.

Lack of Comparison with Latest LiteratureMost references are from before 2024, and no studies on paclitaxel resistance biomarkers in ovarian cancer published in 2024–2025 were included. It is recommended to add a “Comparison with Latest Research” section in the “Discussion” to analyze the similarities and differences between the findings of this study and the latest advances.

In our response, the discussion section now includes the following paragraph:

“Recent studies have identified potential mechanisms for the development of paclitaxel resistance involving TGFB pathways in ovarian cancer cell lines via [85] and cyclin dependent kinase 14 (CDK14) [86] mediated mechanisms.  We applied a multivariate Cox proportional hazards model to investigate the impact of Taxol-treated patients on HR for the prognostic markers identified in our screen.  These analyses showed that statistical significance of the Taxol parameter could not be achieved and that would have warranted further characterization of paclitaxel-sensitive versus paclitaxel-resistant tumors in the TCGA dataset. Likewise, we also tested the impact of bevacizumab on HR in ovarian cancer patients in the TCGA dataset because of the recent introduction of bevacizumab into first-line therapeutic regimens [87] and improvements observed in PFS during second-line maintenance [88]. We observed improvements in OS for 29 patients treated with bevacizumab that did not achieve statistical significance, so we could not further investigate the prognostic impact of the markers described in our study.” (Lines 510-522)